1 Comparative study of flood projections under the climate

# 2 scenarios: links with sampling schemes, probability

- 3 distribution models, and return level concepts
- <sup>4</sup> Lingqi Li<sup>1</sup>, Lihua Xiong<sup>1,\*</sup>, Chong-Yu Xu<sup>1,2</sup>, Shenglian Guo<sup>1</sup>, Pan Liu<sup>1</sup>
- <sup>1</sup>State Key Laboratory of Water Resources and Hydropower Engineering Science, Wuhan University,
   Wuhan 430072, China
- $\frac{1}{2}$  wuhan 450072, China
- 7 <sup>2</sup>Department of Geosciences, University of Oslo, PO Box 1047 Blindern, N-0316 Oslo, Norway
- 8
- 9 E-mail addresses:
- 10 L. Li (lqli@whu.edu.cn),
- 11 L. Xiong (xionglh@whu.edu.cn),
- C.-Y. Xu (c.y.xu@geo.uio.no),
- S. Guo (slguo@whu.edu.cn),
- P. Liu (liupan@whu.edu.cn)
- Correspondence to:
- Lihua Xiong, PhD, Professor
- State Key Laboratory of Water Resources and Hydropower Engineering Science
- Wuhan University
- Wuhan 430072, China
- E-mail: xionglh@whu.edu.cn
- Telephone: +86-13871078660
- Fax: +86-27-68773568

# 24 Abstract

Traditional stationarity strategy for extrapolating future design floods requires 25 26 renovation in response to the possible nonstationarity caused by changing climate. 27 Capable of tackling such problem, the expected-number-of-events (ENE) method is 28 employed with both Annual Maximum (AM) and Peaks over Threshold (POT) 29 sampling schemes expatiated. The existing paradigms of the ENE method are 30 extended focusing on the over-dispersion emerged in POT arrival rate, for which by 31 virtue of the ability to account, the Negative Binomial (NB) distribution is proposed 32 as an alternative since the common assumption of homogeneous Poisson process 33 would likely be invalid under nonstationarity. Flood return levels are estimated and 34 compared under future climate scenarios (embodied by the two covariates of 35 precipitation and air temperature) using the ENE method for both sampling schemes in the Weihe basin, China. To further understand how flood estimation responds to 36 37 climate change, a global sensitivity analysis is performed. It is found that design 38 floods dependent on nonstationarity are usually but not necessarily more different 39 from those analyzed by stationarity strategy due to the interaction between air 40 temperature and precipitation. In general, a large decrease in flood projection could be 41 induced under nonstationarity if air temperature presents dramatically increasing trend 42 or reduction occurs in precipitation, and vice versa. AM-based flood projections are 43 mostly smaller than POT estimations (unless a low threshold is assumed) and more sensitive to changing climate. The outcome of the biased flood estimates resulting 44 45 from an unrestricted use of the Poisson assumption suggests a priority to the NB

- distribution when fitting POT arrival rate with significantly larger variance than the
- mean. The study supplements the knowledge of future design floods under changing
- climate and makes an effort to improve guidance of choices in flood inference.
- Keywords: Nonstationarity; Flood return level; Peaks over Threshold; Annual
- Maximum; Negative Binomial distribution; Climate change

# 52 1. Introduction

Flood frequency analysis, one of the most widely used tools in hydrology, is of great significance for theoretical research and practical application in flood projection and risk management. Reliable flood return level estimation requires careful considerations basically from three aspects, i.e., sampling schemes, probability distribution models, and return level concept.

Primarily, two kinds of sampling schemes are used in common for the flood-related 59 studies (Coles, 2001), i.e., the Annual Maximum (AM) (block defined as year scale in 60 block maxima sampling) and Peaks over Threshold (POT) (also known as partial 61 duration series). The AM sampling, extracting the annual maximum peak flows from 62 the observed discharge series, is simpler than the POT sampling that collects the discharges above a fixed high threshold. Hence, the AM realizes a wider use in 63 hydrology than the POT, but losing 'real flood' information is inevitable because 64 65 small discharge included in a dry year could be misleading (Lang et al., 1999). The 66 POT, free from the sampling restriction of the AM that picks only one event per year, 67 seems to be rational as it substantially contains two flood characteristics to be 68 portrayed separately: the magnitude and the arrival rate (annual number of 69 exceedances above the threshold) (Ön öz and Bayazit, 2001).

No matter for AM or POT floods, flood frequency analysis has undoubtedly, for a long time, indulged in such a prevailing approach that flood events, subject to the underlying assumptions of being independent and identically distributed (*i.i.d.*), share

the same probability distribution. This description can be epitomized as the 74 stationarity strategy used in traditional flood frequency analysis (Coles, 2001). Since 75 the impacts of climate change on hydrological system have been reported repeatedly 76 (IPCC, 2013), nonstationarity, as a special concept in contrast to stationarity, has 77 generally enjoyed popular supports in academia. A number of researchers have been 78 absorbed in, for instance, revealing the invalidation of stationarity strategy (Khaliq et 79 al., 2006; Milly et al., 2008), describing the temporal variability of hydrological 80 characteristics (Villarini et al., 2009a; Machado et al., 2015; Xiong et al., 2015a), and 81 exploring the reasons behind the changes (Ishak et al., 2013; López and Francés, 2013; 82 Jiang et al., 2015; Xiong et al., 2015b).

Important as it is, questions around "stationarity is still alive or wanted dead" (Lins 84 and Cohn, 2011; Koutsoyiannis, 2011) have been subsequently pointed out sharply, 85 remaining more or less as a controversial puzzle. In an attempt to clarify these issues, 86 so far there have appeared various arguments. For example, Lins and Cohn (2011) 87 admitted the existence of nonstationarity but simultaneously suggested the use of 88 stationarity to elude the potentially high uncertainty of nonstationary influences on 89 hydrologic studies. Montanari and Koutsoyiannis (2014) asserted that stationarity is 90 immortal for the need of mitigating natural hazards. Koutsoyiannis and Montanari 91 (2015) stated, persuasively, that the misunderstanding of stationarity has let "changes" 92 be mistakenly labeled as "nonstationarity." Likewise, Serinaldi and Kilsby (2015) 93 deliberately titled their main topic with "stationarity is undead" to alert of the 94 uncertainty related to nonstationary flood frequency analysis. In response to the

thoughtful literatures with the opposing opinions mentioned above, Milly et al. (2015) reiterated the viewpoints of Milly et al. (2008) who claimed that "stationarity is dead" 96 97 by using "Policy Forum" to communicate the necessity of considering nonstationarity 98 in hydrology in the 21<sup>st</sup> century. Stedinger and Griffis (2011) explained conservatively 99 that formulating nonstationary models with finite flood records can be defensible 100 when physical-causal basis for multi-decadal projections is known. Indeed, 101 stationarity, as the solid cornerstone laid for hydrologic frequency analysis, does 102 deserve to be active (Koutsoyiannis, 2011). Nevertheless, there is reason to afford an 103 opportunity to nonstationarity for advancing hydrologic research (Milly et al., 2015). 104 Advocating nonstationarity at present is intended to arouse the consciousness in the 105 scientific community due to the on-going climate changes yet without smothering 106 stationarity.

In the presence of nonstationarity, a good few of studies (also this paper) 108 materialize nonstationary hydrologic variables with resorting to the time-variant 109 characters of the variable moments (Khaliq et al., 2006), i.e., nonstationary 110 flood-frequency distribution model is constructed by the theoretical probability 111 distribution whose statistical parameters are assumed to be no longer fixed over time 112 (Stedinger and Griffis, 2011; Milly et al., 2015). To addressing the causes of 113 nonstationarity, researchers attribute the changes, qualitatively, by nonparametric 114 cross-correlation analyses (Ishak et al., 2013), and quantitatively, by linking the 115 time-varying distribution parameters to the exploratory variables, e.g., time and 116 potential driving forces (Prosdocimi et al., 2015; Serinaldi and Kilsby, 2015; Silva et

al., 2015; Xiong et al., 2015b), among many others.

| 118 | Analyses for flood frequency or return level have been accomplished with the AM,       |
|-----|----------------------------------------------------------------------------------------|
| 119 | POT, or both worldwide with either stationary or nonstationary hypotheses (e.g.,       |
| 120 | Villarini et al., 2012; López and Francés, 2013; Machado et al., 2015; Xiong et al.,   |
| 121 | 2015a). However, attentions paid to the comparison of AM and POT flood series in       |
| 122 | flood frequency analysis are relatively limited in the previous research endeavors.    |
| 123 | Rosbjerg (1985) whose research was completed on a stationary background deemed         |
| 124 | that the POT series modeled with heavy-tailed distributions should yield more          |
| 125 | advisable flood estimates than AM. Madsen et al. (1997) suggested that the POT         |
| 126 | series was generally preferable to AM series for at-site flood estimation under        |
| 127 | stationarity. More recently, Bezak et al. (2014) found that the POT series gave higher |
| 128 | flood estimates than the AM series for larger return periods on stationary conditions. |
| 129 | Prosdocimi et al. (2015) concluded that POT models outperformed AM models in           |
| 130 | respect of detecting the external causes of nonstationary floods.                      |

It is worth noting that nonstationarity in the flood series caused by the changing 132 environments has made stationarity strategy for return level estimation problematic 133 (Khaliq et al., 2006; Sivapalan and Samuel, 2009; Villarini et al., 2009b; López and 134 Francés, 2013). There have been numerous studies on return level inference 135 associated with hydro-climatic extreme events that consider nonstationary conditions. 136 For example, return level was proposed, with the corresponding return period as the 137 expected waiting time until an exceedance occurs (Olsen et al., 1998; Wigley, 2009; 138 Salas and Obeysekera, 2014), or as the quantile over which the expected number of

events (ENE) during a given return period is one (Parey et al., 2007, 2010).
Comparative analyses on such two methods were performed in Cooley (2013), Du et
al. (2015), etc. Besides, risk-oriented approaches for deriving flood estimators
considering nonstationarity have also been devised in some literatures with diversities
in their scope (Sivapalan and Samuel, 2009).

These profound studies, with considerable efforts made on the extrapolation of 145 hydro-climatic extremes under nonstationarity, have presented capacity and depth in 146 theory, and most focus on the block (e.g., AM) sampling. Other sampling, i.e., the 147 POT, seems not to receive much attention in estimating return levels by the method 148 adapted to the context of nonstationarity in the literatures except Parey et al. (2010) 149 who set an example with the application of the ENE method to the POT case yet 150 without much more discussions on mathematical treatment, and Silva et al. (2015) 151 who estimated the flood hazards based on the POT framework by making the 152 engineering design life period equal to the past observation periods. Additionally, 153 exploration on future design floods in nonstationarity context is still limited as well as 154 the analyses on how climate change could influence flood projections.

This paper is aimed to achieve multi-decadal flood projections under the future climate scenarios and investigate the effect of climate changes on design floods. Essentially, the study can serve as a complement of the available ENE method from the following aspects. First, design floods are estimated with two sampling schemes of AM and POT and compared on not only stationary but also nonstationary conditions. Second, the ENE method is extended for the POT sampling with an

161 emphasis on describing the POT arrival rates. The POT arrival rates have in fact been 162 chronically accepted on faith to follow a homogeneous Poisson process under 163 stationarity (Shane and Lynn, 1964). However, such assumption has been reported to 164 be invalid due to two-type sources of nonstationarity which will be addressed herein: 165 (i) heterogeneity of Poisson process intensity (Cunnane, 1979; Villarini et al., 2012; 166 Silva et al., 2015), for which the Poisson distribution is retained no longer with 167 invariant Poisson process intensity, or rather, parameterized as functions of climatic 168 covariates; (ii) over-dispersion of observations. Theoretically, the Poisson distribution 169 holds identical variance and mean of population, whereas it is often the case that the 170 variance is rarely equal to, and even significantly higher than, the mean (Cunnane, 171 1979). Therefore, the Negative Binomial (NB) distribution is recruited as an 172 alternative to the Poisson distribution following the findings from Ben-Zvi (1991) and 173 Önöz and Bayazit (2001). Finally, the sensitivity of flood estimations to changing 174 climate is analyzed for reference to future inference.

# 175 **2. Methodology**

Analysis of flood return levels is undertaken briefly following: preliminary diagnosis for nonstationarity evidence, modeling of both AM and POT samplings under stationarity and nonstationarity, respectively, (i.e., using the assumed probability distributions with parameters as functions of constant or climatic covariates), extrapolation of flood by applying the ENE method to these models, and investigation on how climatic effect affects flood estimations.

### 182 **2.1. Diagnostics for nonstationarity**

Justifying the presence of nonstationarity is of great importance for the investigation 184 of hydro-climatic events in a changing world (Montanari and Koutsoyiannis, 2014; 185 Serinaldi and Kilsby, 2015; Milly et al., 2015; Xiong et al., 2015b). Importance 186 attached to the gradual evolution of observation time series, is emphasized, for which 187 the preliminary detection is implemented by three nonparametric trend tests: the 188 Mann-Kendall (MK) (Mann, 1945; Kendall, 1975), the pre-whitening (PW) (von 189 Storch, 1995), and the trend-free pre-whitening (TFPW) (Yue et al., 2002). The latter 190 two tests are proposed initially to mitigate the adverse influence of lag-1 serial 191 correlation  $r_1$  on the robustness of the MK method. Instead of testing the MK 192 statistics  $Z_{MK}(\cdot)$  of the original observation series  $\{X_t, t = 1, 2, ..., N\}$ , they use the 193 new independent series of  $X'_{t} = X_{t} - r_{1}X_{t-1}$  and  $Y''_{t}$  from Eq. (1), respectively.

$$S = \operatorname{median}_{\forall t_{1} 

testing the significance of trend in the modified dependent variable after removing the 202 linear dependence on a covariate. It is inferred that dependent variable may co-vary 203 with the physical covariate if the *p*-value of  $Z_{PMK}$  becomes larger than the given 204 significance level (0.05). The closer the *p*-value is to one, the greater the extent to 205 which the dependent variable relates to the physical covariate.

To verify the conjecture if the homogeneous Poisson process assumption is valid 207 under changing circumstances, the Bohning (1994) test is applied to the observed 208 series of POT arrival rates for testing against the alternative hypothesis that the variance of population  $S^2$  is larger than the mean  $\overline{X}$ . The test statistic 209  $\sqrt{\left(\frac{n-1}{2}\right)\left(\frac{S^2}{\overline{X}}-1\right)}$  asymptotically converges to the normal distribution for a large 210 211 population. Given the finite sample size, a bootstrap simulation is performed to 212 generate randomly 10000 replications from original series and for each replication 213 calculate the statistic values. According to the given significance level (0.05), the 214 Poisson assumption would be rejected if the *p*-value of the attained empirical 215 distribution for test statistic is less than 0.05.

### 216 2.2. Probability distribution modeling

Modelling of flood series was undertaken for recruiting the theoretical probability distribution as potential candidates. In this paper, the distribution to be considered is selected based on the successful applications in previous studies (e.g., Madsen et al., 1997; Lang et al., 1999; Du et al., 2015) but for the purpose at current stage not including all of them. The AM floods are assumed to follow three different types of