# Peer review of "scenarios: links with sampling schemes, probability"

_Hydrology and Earth System Sciences, 2016_

## Referee Comment (RC1) · S. Parey (Referee) · 22 Dec 2016

The paper discusses the estimation of flood return levels in the nonstationary context and applies the ENE (Expected Number of Exceedances) concept for both block maxima and Peak Over Thresholds approaches. It is an interesting, well presented and documented study using pertinent methodologies. The flow data used for the application extend over the period 1960-2009, which leads to 50 annual maxima, which is good but not perfect for a robust statistical fitting. Could you argue why other distributions than the asymptotic limit GEV distributions are considered for fitting annual maxima?

LP3 is pointed as the best choice, but it is the one with the largest standard errors on the parameter estimations in the stationarity case Concerning the POT approach, in theory, under stationary conditions, the return levels should be the same whatever the approach used to estimate them, which is the case here if we compare POT2 and GEV with their confidence intervals (which are larger for GEV because the fitting is made with less values). I have questions concerning the POT approach: Threshold choice: there are some rules to choose a convenient threshold for the POT estimation, based on the mean excess plot and/or the constancy of the shape and modified scale parameters. Considering the very different results obtained with POT4 compared to POT2 or 3 (and GEV), it is doubtful that the threshold used for POT4 is a convenient threshold Seasonality: the threshold exceedances have to be independent (which is correctly dealt with in the excesses selection) and identically distributed. This second condition is not considered here, but may not be straightforward for environmental variables, because of seasonality and possibly inter-annual variability. Regardless of inter-annual variability, is there a preferred season for the occurrence of floods in this basin? If so, then it may be necessary to restrict the analysis to this season. This has an impact on the estimated Poisson intensity. The Poisson process however does not need to be homogeneous, when nonstationarity is introduced, it is a non-homogeneous Poisson process. Arrival rate: the use of a Negative Binomial is interesting, but it brings one more parameter to estimate with still quite few values. Could you discriminate the advantage brought by this approach compared to this necessity to estimate another parameter? Then the sensitivity analysis is very interesting, as well as the variations in return levels induced by the separate increases of Ptotal and Tmean. It could be much more informative if a choice had been done previously of the best model for the estimation.

---

## Referee Comment (RC2) · F. Serinaldi (Referee) · 3 Jan 2017

**General comments**

In this study, the Authors perform quite a standard nonstationary frequency analysis. It can seem surprising to talk about "standard analysis" when dealing with relatively new "nonstationary" fashion, but the main point is that this paper, as a large part of those on this topic, is a simple application of a set of routines already implemented in R packages. Thus, most of the results are quite speculative, as they overlook the theoretical

concepts behind statistical tools and some significant papers explaining these issues.

Moreover, I regret to say that a very similar version of this paper was already declined by Advances in Water Resources in 2015. In that case, I suggested a major revision, but I see that the Authors did not account for most of my suggestions. Some of them are reported below once again with updates.

The main contribution of this study should be the use of the expected-number-of-events (ENE) method and the negative binomial distribution to replace the Poisson distribution under overdispersion conditions. Concerning the ENE method, it is only one of the possible approaches to define return periods and corresponding return levels. It yields the general equation $T = \frac{T}{m \sum_{t=1}^{T}(1-F_t(x_T))} = \frac{1}{mE[1-F_t(x_T)]}$, which reduces to the classical $T = \frac{1}{m(1-F(x_T))}$ if $F_t(x_T) \equiv F(x_T)$ for each $t$. In other words, the return period corresponds the expected value of the reciprocal of the exceedance probability of a fixed quantile $x_T$, which is constant if $F_t(x_T)$ is constant. Therefore, sentences such as "This advantage makes the method able to provide unique design value for reference even though the flood behaviors observe nonstationarity, which is beyond the capacity of traditional stationarity strategy" make little sense because return periods and return levels, being expected values taken over $T$ (for ENE) or $\infty$ (for expected waiting time), are always unique values in both stationary and nonstationarity framewoks (a discussion is provided by Serinaldi (2015)). As far as the negative binomial distribution is concerned, it was already discussed in stationary flood frequency analysis, and compared with Poisson by Bhunya et al. (2013), while its introduction and theoretical justification in stationary and nonstationarity frameworks was presented by Eastoe and Tawn (2010). In particular, the latter highlighted how the overdispersion is not necessarily a consequence of nonstationarity. In fact, overdispersion can easily results from (hidden) persistence (see e.g. Serinaldi F, Kilsby CG. (2016a) and references therein) and/or mixed effects (random fluctuations of the rate of occurrence). Moreover, saying that "over-dispersion of observations" is a possible source of nonstionarity (P9L161-168) seems to me logically flawed because overdispersion is not a cause but an effect

[Figure]

of non-Poissonian behaviour, which can have many different causes. Moreover, other models such as generalized Poisson can be used (Raschke, 2015). Again, nonstionarity is not a necessary condition. However, what really matters is that the distribution of the number of event under nonstationarity is neither Poisson nor negative binomial, but Poisson binomial (Tejada and den Dekker, 2011; Obeysekera and Salas, 2016). Thus, a more careful literature review should be performed before running (a bit blindly) computer codes/packages.

Most of the conclusions in the case study are quite speculative because the behaviour of the return periods under nonstationarity depends on many factors, such as the link functions, the relationships between distributions' parameters and covariates (linear or polynomial regression are surely convenient but also quite arbitrary and surely not physically based), as well as the nature of the distributions (fat tailed, heavy tailed, etc.). In this respect, conclusions are quite fair as they reflect the overall uncertainty of the empirical results, which is however exacerbated by lack of theoretical reasoning on the rationale and true nature of the methods used. By the way, it is worth noting that the models with parameters depending on covariates that exhibit stochastic fluctuations (such as rainfall and temperature) are not nonstationary but simply doubly stochastic. Nonstationary models require that the distribution (marginal and joint) change $with time$ according to some well-defined function holding true for whatever instant along the time axis. In this respect, trend analysis can only detect local changes in a very small interval (i.e., the period of record), and this explains why nonstationarity cannot be inferred from trend analysis but requires exogenous information, i.e. attribution based on physical reasoning rather than statistical correlation analysis.

Some additional specific remarks are provided below. I also refer the Authors to my report for the previous AWR version.

**Specific comments**

L141-143: see Serinaldi (2015) for a wider discussion.

L146: "other sampling methods. . .seem"

L189: TFPW does not perform any prewhitening and does not preserve the nominal significance level. This explains why the results reported in the literature for MK and TFPW MK are often close to each other (see Serinaldi and Kilsby (2016b) for an analytical and numerical proof)

L202-205: The interpretation of the partial MK test is not correct. Moreover, the interpretation reflects a widespread merging of Neyman-Pearson testing procedure and Fisher's p-values, whose values cannot be interpreted as proxies of the strength of the relationship between target variables and covariates.

L268-275: AIC and BIC have different meaning and are relative measures. Thus, model selection should be based on Akaike weights and/or evidence ratios (see Burnham and Anderson (2002,2004)).

L289: If the process is not stationary, the empirical frequencies cannot be computed by the Gringorten formula. Classical qq plot does not make sense in a (true) nonstationary framework. Moreover, in GAMLSS, the residuals are not the normal quantile transform of the observed values (this holds only for stationary models) but the difference between the predictions (given the covariates) on the observations (for the same covariates). Furthermore, qq plots and coefficient of determination are not formal tests but diagnostic plots and measures of performance, respectively. In particular, no tests can be performed at the 5% significance level for coefficients of determination (unless ad hoc MC experiments are set up).

L494-496: This result is not so surprising because the number of exceedances decreases as the threshold increases, and therefore the clustering of extreme events is more evident given the short time series.

Section 4.3: GAMLSS are nothing but an advanced form of regression. Using the fitted model with covariates taking values beyond the fitting range is never a good idea because we do not know if the fitted relationship still holds true in that range.

**Editing remarks**

English should be revised and some typos fixed.

Sincerely,
Francesco Serinaldi

**References**

Bhunya P.K., R. Berndtsson, S. K. Jain, R. Kumar (2013) Flood analysis using negative binomial and Generalized Pareto models in partial duration series (PDS), Journal of Hydrology, 497, 121-132.

Burnham K. P. and Anderson D. R. (2002) Model Selection and Multimodel Inference: A Practical Information-Theoretic Approach, Springer

Burnham K. P. and Anderson D. R. (2004) Multimodel Inference: Understanding AIC and BIC in Model Selection, Sociological Methods Research 2004; 33; 261

Eastoe, E. F., and J. A. Tawn (2010), Statistical models for overdispersion in the frequency of peaks over threshold data for a flow series, Water Resour. Res., 46, W02510, doi:10.1029/2009WR007757.

Obeysekera, J. and J. D. Salas (2016) Frequency of Recurrent Extremes under Non-stationarity, Journal of Hydrologic Engineering, 21(5), doi: 10.1061/(ASCE)HE.1943-5584.0001339

Raschke, M. (2015) Statistical detection and modeling of the over-dispersion of winter storm occurrence, Nat. Hazards Earth Syst. Sci., 15, 1757-1761.

Serinaldi F. Dismissing return periods! (2015). Stochastic Environmental Research and Risk Assessment, 29(4), 1179-1189

Serinaldi F, Kilsby CG. (2016a) Understanding persistence to avoid underestimation of collective flood risk. Water, 8(4), 152.

Serinaldi F, Kilsby CG. (2016b)The importance of prewhitening in change point analysis under persistence. Stochastic Environmental Research and Risk Assessment, 30(2), 763-777

Tejada, A., and den Dekker, A. J. (2011). The role of Poisson's binomial distribution in the analysis of TEM images. Ultramicroscopy, 111(11), 1553–1556.
* * *

---

## Referee Comment (RC3) · Anonymous Referee #3 · 4 Jan 2017

General comments:

In the submitted paper authors compare different methods that can be selected to perform the flood frequency analysis (FFA). Both stationary annual maximum (AM) method and peaks over threshold (POT) method are tested using the daily discharge data from the Huaxian station (Weihe basin) in China. Further, a nonstationary methodology is also applied and compared with the stationary approach. Several interesting and important aspects of the flood frequency analysis approach are analyzed and discussed: application of the expected-number-of-events (ENE) method, selection of different distribution functions in the AM method, evaluation of the suitability of the Poisson and negative binomial (NB) distributions to model the annual number of exceedances above the threshold, comparison between the stationary and nonstationary approaches using both AM and POT methods, sensitivity analysis regarding the influence of the precipitation and temperature on the nonstationary flood frequency analysis results.

The paper is relatively well written and the presented topic is interesting for the hydrological society due to the importance of the flood frequency approach for the design of different hydro-technical structures. The paper is in the scope of the journal. However, I would suggest that authors try to put more focus on next points related to the practical applications of the FFA, because the presented paper does not develop a new theory but compares different aspects of the FFA approach:

1) The authors have performed detailed analysis and comparison of different methods that can be used to carry out the flood frequency analysis. Is it possible to point out which method should be used by practitioners to determine the design flood (taking into account the larger sensitivity of the nonstationary AM method compared to the POT, more complicated POT analysis compared to the AM method and other conclusions stressed in this paper)? Should the standard procedures to perform the FFA in China be modified after the results of this study? What is the trade-off (if any) between the model complexity and uncertainty in the flood frequency analysis results?

2) Related to the previous point, different methods yielded diverse FFA results. For example, the 50-year flood was estimated to be between approximately 4000 and 8000 m3/s with the consideration of the confidence intervals (Fig. 5). Can the authors suggest some guidance for selection of the most appropriate method to carry of the FFA?

3) Looking at the results of the nonstationary approach (AM method) shown in Fig. 5 it seems that the return level increases to about 30-year return period and then it is almost constant for larger return periods? Does this means that the 50-year flood is the same as the e.g. 200-year flood? Please explain.
4) It would be interesting to make a comparison of the nonstationary approach where the model parameters change with time (e.g., Obeysekera and Salas, 2014; Salas and Obeysekera, 2014; Sraj et al., 2016; Vogel et al., 2011) and not only with P and T.

The English is understandable, but it could benefit from some improvements, therefore I recommend editing for English language.

Specific comments and technical corrections:

Page 13, line 257: I would suggest adding a reference for the GAMLSS package.

Page 16, lines 313-314: I would suggest rephrasing this sentence.

Page 16, line 321: What is "dramatic" or "pointless" for the authors? This can be very subjective, thus I would suggest avoiding such statements.

Page 19, line 387: Which Sensitivity package (a reference should be added)?

Page 21, line 408: Replace "134,800" with "134 800" (and also in some other parts of the manuscript).

Page 21, line 414: Upstream and not downstream?

Page 22, line 422: Replace "Thiessen polygon" with "Thiessen polygons".

Page 24, lines 487-488: Any particular physical reason for this negative trend? It would be interesting to see the discharge data used in study.

Page 24, line 492: Again, what does "dramatically" means?

Page 25, lines 499-503: Is this the case for all 22 analyzed stations?

Page 26, lines 526-529: I would suggest rephrasing this sentence.

Page 26, line 537: "much lower" this is subjective; I would suggest using the % to show the difference.

Page 29, lines 569-570: What is reason for this large difference and what does this

mean from the perspective of the practitioners?

Page 29, lines 583-585: These are relatively large differences. Which POT threshold is suggested by the authors and why?

Page 30, line 618: Dot is missing at the end of the sentence.

Page 31, line 642: Replace "shows" with "show".

Page 32, line 652: "there is not much difference" looking at Fig. 5 I would say that differences are relatively large for some cases?

Page 33, line 672: Replace "if we allowing" with "if we are allowing".

Page 33, line 679: Replace "requires" with "require".

Page 36, line 748: Reason for this difference?

Page 36, lines 748-751: What does this conclusion means for the practical application of the FFA?

Page 37, lines 760-763: This is very important conclusion but is it true only for this case study or there is a theoretical background for it?

Page 39, lines 807-810: But this "relatively complicated sampling criteria" still exists and if we compare the POT sampling methodology with the nonstationary approach used in this study I would say that it is even more complicated (than the stationary approach) and it requires additional knowledge?

Page 40, 820-823: What does this means from the practical perspective?

Page 40, lines 830-832: Does this hold for this case study or in general?

References:

Obeysekera, J., Salas, J.D. 2014. Quantifying the uncertainty of design floods under nonstationary conditions. Journal of Hydrologic Engineering, 19, 1438–1446.

Salas, J.D., Obeysekera, J. 2014. Revisiting the concepts of return period and risk for nonstationary hydrologic extreme events. Journal of Hydrologic Engineering, 19, 554–568.

Sraj, M., Viglione, A., Parajka, J., Blöschl, G. 2016. The influence of non-stationarity in extreme hydrological events on flood frequency estimation. Journal of Hydrology and Hydromechanics, 64, 426–437.

Vogel, R.M., Yaindl, C., Walter, M. 2011. Nonstationarity: Flood magnification and recurrence reduction factors in the United States. Journal of American Water Resources Association, 47, 464–474.

---

## Author Comment (AC1) · 2 Mar 2017

**Responses to Referee #1**

**Dear Referee #1,**

We really appreciate your rapid and constructive comments on our manuscript entitled "Comparative study of flood projections under the climate scenarios: links with sampling schemes, probability distribution models, and return level concepts" (Number: hess-2016-619) that are very helpful to improve our study and paper. We have carefully followed these comments and accordingly made the revisions. Please see our point-by-point reply below.

**General comment 1:**

The paper discusses the estimation of flood return levels in the nonstationary context and applies the ENE (Expected Number of Exceedances) concept for both block maxima and Peak Over Thresholds approaches. It is an interesting, well presented and documented study using pertinent methodologies. The flow data used for the application extend over the period 1960-2009, which leads to 50 annual maxima, which is good but not perfect for a robust statistical fitting.

**Response:**

We thank the referee very much for the positive evaluation of our paper and the valuable comments. Indeed, a long-period dataset is essential for a robust statistical fitting as the longer series are considered to better represent the population. Nevertheless, a compromise has to be made for a real-world application since long-historical records are often sparse due to accidental (e.g., equipment malfunction) or human-induced reasons (e.g., management failure). This is why even shorter data series are used in many other studies, e.g., Villarini et al. (2009) used 26 annual flood peak of the Little Sugar Creek watershed in Charlotte, North Carolina to fit a gamma distribution model under nonstationarity. Seckin et al. (2011) conducted flood frequency analysis with the annual flood peak records varying in lengths from 15 to 57 years.

Following the referee's suggestion and in order to further improve the statistical practice, we have attempted to collect more available data for our study, which is now extended to 2012, thus the study period is from 1951 to 2012 (totally 62 years). Due to the augmentation of data for the study, the related statements, figures and tables in the data description and result analysis sections have been changed accordingly in the revised manuscript.

Reference

Villarini, G., Smith, J.A., Serinaldi, F., Bales, J., Bates, P.D., and Krajewski, W.F.: Flood frequency analysis for nonstationary annual peak records in an urban drainage basin, Adv. Water Resour., 32, 1255-1266, doi:10.1016/j.advwatres.2009.05.003, 2009.

Seckin, N., Haktanir, T., and Yurtal, R.: Flood frequency analysis of Turkey using L-moments method, Hydrol. Process., 25, 3499-3505, doi:10.1002/hyp.8077, 2011.

**General comment 2:**

Could you argue why other distributions than the asymptotic limit GEV distributions are considered for fitting annual maxima? LP3 is pointed as the best choice, but it is the one with the largest standard errors on the parameter estimations in the stationarity case.

**Response:**

Thank you for the valuable comments. In many available literatures, flood frequency analysis for annual maxima (AM) data has been frequently carried out with several theoretical distributions selected from the Normal family (e.g., normal, lognormal), the Gamma family (e.g., gamma, Pearson type 3, Log-Pearson type3), and the Extreme Values family (e.g., Weibull, Gumbel, Generalized Pareto) since there is no conclusive standard to define the adoption of a single type of optimum distribution for AM series. All the distributions presented above have been widely used, and their respective advantages found at different gauging sites and/or in different river basins have been proven (Strupczewski et al., 2001; Kidson and Richards, 2005; Villarini et al., 2009). The asymptotic limit GEV distribution is a theoretical result based on Extreme Values Theory and may not be always applicable under the complex hydrological circumstances and/or with the finite observation data. Additionally, distributions recommended for flood frequency analysis by national standards in many countries are often different, e.g., LP3 distribution is officially specified in both USA and Australia (Vogel et al., 1993), while P3/LP3 is recommended in China. Therefore, except for the GEV distribution, we have employed other distributions (i.e., LNO3, LP3) from two different distribution families in this paper. These three distributions all have the parameters of location, scale, and shape, which we consider can allow a flexible fit for annual maxima data. To better account for the choice of distributions for AM series, we have clarified these points in the newly revised manuscript.

The model selection follows a generalized Akaike information criterion, i.e., AIC and BIC, to make a tradeoff between model structure complexity and goodness of fit. The model with minimum generalized Akaike information criterion value is preferentially considered, and it will be finally chosen as the best once if the significance test of parameter estimations and goodness-of-fit test can pass at the 5% significance level. From this, we can see that the

selected best model does not necessarily mean a lowest standard error for parameter estimations (due to the uncertainty problem incurred by limited observation data). As the referee pointed out, this study indeed has found that the LP3 model yields a larger standard error (but the difference is minor) in the estimated statistical parameters than other used probability distribution models, including GEV, in the stationary context. However, this LP3 model does have the minimum AIC/BIC value when compared to other used probability distribution models such as GEV, and has passed the 5% significance test for statistical parameters.

Reference

Kidson, R., and Richards, K.S.: Flood frequency analysis: assumptions and alternatives, Prog. Phys. Geog., 29, 392-410, doi:10.1191/0309133305pp454ra, 2005.

Strupczewski, W.G., Singh, V.P., and Mitosek, H.T.: Non-stationary approach to at-site flood frequency modelling. III. Flood analysis of Polish rivers, J. Hydrol., 248, 152-167, doi:10.1016/S0022-1694(01)00399-7, 2001.

Villarini, G., Smith, J.A., Serinaldi, F., Bales, J., Bates, P.D., and Krajewski, W.F.: Flood frequency analysis for nonstationary annual peak records in an urban drainage basin, Adv. Water Resour., 32, 1255-1266, doi:10.1016/j.advwatres.2009.05.003, 2009.

Vogel, R.M., McMahon, T.A., and Chiew, F.H.S.: Flood flow frequency model selection in Australia, J. Hydrol., 146, 421-449, doi:10.1016/0022-1694(93)90288-K, 1993.

**General comment 3:**

Concerning the POT approach, in theory, under stationary conditions, the return levels should be the same whatever the approach used to estimate them, which is the case here if we compare POT2 and GEV with their confidence intervals (which are larger for GEV because the fitting is made with less values). I have questions concerning the POT approach.

**3.1** Threshold choice: there are some rules to choose a convenient threshold for the POT estimation, based on the mean excess plot and/or the constancy of the shape and modified scale parameters. Considering the very different results obtained with POT4 compared to POT 2 or 3 (and GEV), it is doubtful that the threshold used for POT4 is a convenient threshold.

**Response:**

Thanks for this comment and suggestion. In the original paper, we followed the criteria in USWRC (1982) to define time span between successive extreme events, which guarantees the

independence of the selected peak flow values. We then choose the threshold value *u* for POT flood samples according to a widely-used method proposed in Lang et al. (1999), i.e., the preselected annual number of peaks per year on average. For example, POT2 threshold is determined based on that the number of selected peaks is twice the number of years of the study record. This procedure for threshold choice includes somewhat a level of subjectivity since there is no absolutely the best approach to determine the threshold, and no unique value that must be selected but rather a range of appropriate values. Considering this lack of standard methods in determining the threshold, we apply three different threshold values to define the POT samples for our study objectives and want to see if the results are dependent on the threshold choice.

In accordance with the referee's comment and suggestion, we have added in the newly revised manuscript the tests, namely, the mean excess plot and the plot for estimated shape and scale parameters, to evaluate the reasonability of the selected POT samples. In fact, the three used threshold values are all tested to be acceptable, but the use of POT4 as the referee pointed out may not be the most appropriate choice for flood design in the study basin of the Weihe. For example, the results of the mean excess plot indicate that the POT4 threshold almost approaches (but is within) the lower bound of domain where the mean excess should be an approximately linear function of *u* for a valid choice of the generalized Pareto (GP) distribution.

Reference

Lang, M., Ouarda, T.B.M.J., and Bobée, B.: Towards operational guidelines for over-threshold modeling, J. Hydrol., 225, 103-117, doi:10.1016/S0022-1694(99)00167-5, 1999.

USWRC: Guidelines for Determining Flood Flow Frequency, United States Water Resources Committee, Washington, DC, USA, 1982.

**3.2** Seasonality: the threshold exceedances have to be independent (which is correctly dealt with in the excesses selection) and identically distributed. This second condition is not considered here, but may not be straightforward for environmental variables, because of seasonality and possibly inter-annual variability. Regardless of inter-annual variability, is there a preferred season for the occurrence of floods in this basin? If so, then it may be necessary to restrict the analysis to this season. This has an impact on the estimated Poisson intensity. The Poisson process however does not need to be homogeneous, when nonstationarity is introduced, it is a non-homogeneous Poisson process.

**Response:**

Thanks so much for this constructive comment. The seasonal (intra-annual) variability of POT flood, as the referee stated, does exert an important effect on the assumption of Poisson process, which however was not taken into account before. In the new version, we have supplemented the analysis of the distribution and strength of the seasonality based on the longer flow record (as explained in the response to General comment 1) in the Weihe basin, China. The result shown as circular data (Pewsey et al., 2013) in the Figure 1 indicates that most flood events tend to occur during the July-October period with an approximately unimodal distribution. The above information indicates that the intra-annual variability of POT arrival rate is not prominent in this study. Choosing a whole year or a restricted season will lead to the same results.

[Figure]

Figure 1 Flood events (blue points) of the POT2, POT3, and POT4 shown on the circular time axis. The red solid arrow is the mean resultant vector indicating the average occurrence time of the events.

Reference

Pewsey, A., Neuhäuser, M., and Ruxton, G. D.: Circular statistics in R. Oxford University Press, 2013.

**3.3** Arrival rate: the use of a Negative Binomial is interesting, but it brings one more parameter to estimate with still quite few values. Could you discriminate the advantage brought by this approach compared to this necessity to estimate another parameter?

**Response:**

We thank the referee's approval on our work and the good suggestion. To address the question raised by the referee, we firstly recall the traditional use of the Poisson distribution in the POT analysis. The Poisson distribution has the single parameter termed Poisson process intensity. It is characterized by the identity of variance and mean of population, both of which equal to the Poisson process intensity. The POT arrival rate can be fitted by a stationary Poisson model (with constant Poisson process intensity) if it is independent and identically

distributed and follows a homogeneous Poisson process, while a nonstationary Poisson model with time-varying Poisson process intensity can be assumed if the POT arrival rate is independent but not identically distributed and follows a nonhomogeneous Poisson process.

The attractiveness of the use of Negative Binomial (NB) distribution in contrast with the Poisson distribution is explained as follows:

(1) The derivation of the NB distribution is theoretically an extension of the Poisson distribution that mixes Poisson process intensity with a gamma distribution (Anscombe, 1950) (Please refer to Table 1 for the probability mass function in the original manuscript), i.e., the Poisson distribution is a special case of the NB distribution.

(2) The assumption of Poisson distribution is invalid when data is over-dispersed (variance-to-mean ratio greater than unity), while the NB distribution is theoretically justifiable for describing over-dispersed data and has been frequently used in literatures (e.g., Bhunya et al., 2013). Most studies have applied a stationary NB model (with constant parameters) to fit the over-dispersed data, but only a few ones have focused on evaluating whether the over-dispersed data have also shown a nonstationary behavior over a certain long time period. Therefore, this study has been partially aimed to compare the accuracy of stationary and nonstationary NB models for fitting the over-dispersed data.

(3) The requirement of independent data is relaxed when we fit a nonstationary NB model (which is strictly required by the Poisson model, whether with constant or time-varying Poisson process intensity). This advantage has made the NB distribution become increasingly popular especially when it is doubtful whether the observed arrival rates from a stochastic process satisfy the assumption of independence (Johnson et al., 1992).

Reference

Anscombe, F.J.: Sampling theory of the negative binomial and logarithmic series distributions, Biometrika, 37, 358-382, doi:10.2307/2332388, 1950.

Bhunya, P. K., Berndtsson, R., Jain, S. K., and Kumar, R.: Flood analysis using negative binomial and Generalized Pareto models in partial duration series (PDS), J. Hydrol., 497, 121-132, doi:10.1016/j.jhydrol.2013.05.047, 2013.

Johnson, N. L., Kemp, A. W., and Kotz, S.: Univariate Discrete Distributions, Second Edition, New York, John Wiley and Sons, 1992.

**General comment 4:**

Then the sensitivity analysis is very interesting, as well as the variations in return levels induced by the separate increases of Ptotal and Tmean. It could be much more informative

if a choice had been done previously of the best model for the estimation.

**Response:**

We are pleased that our work can be appreciated by the referee and we would like to thank the referee for the valuable remark. We agree with the reviewer that it could be more informative and more instructive if the best model had been chosen in prior. But, in this study, both the choice of the best model and the sensitivity analysis of flood estimation to changing climate resulted from different models as measured by Ptotal and Tmean are the research topics.

Thanks again for your professional and valuable comments in reviewing our paper.

With best wishes

Yours sincerely

Lihua Xiong, PhD, Professor

State Key Laboratory of Water Resources and Hydropower Engineering Science

Wuhan University

Wuhan 430072, PR China

E-mail: xionglh@whu.edu.cn

Telephone: +86-13871078660

Fax: +86-27-68773568

---

## Author Comment (AC2) · 2 Mar 2017

**Responses to Referee #2**

**Dear Referee #2,**

We thank you very much for the professional and constructive comments on our manuscript entitled "Comparative study of flood projections under the climate scenarios: links with sampling schemes, probability distribution models, and return level concepts" (Number: hess-2016-619). The comments are all significant for improving our research and paper. We have carefully followed these comments and suggestions. The point-by-point responses to the comments are shown below. The corresponding revision to the manuscript would be made.

**General comment 1:**

In this study, the Authors perform quite a standard nonstationary frequency analysis. It can seem surprising to talk about "standard analysis" when dealing with relatively new "nonstationary" fashion, but the main point is that this paper, as a large part of those on this topic, is a simple application of a set of routines already implemented in R packages. Thus, most of the results are quite speculative, as they overlook the theoretical concepts behind statistical tools and some significant papers explaining these issues. Moreover, I regret to say that a very similar version of this paper was already declined by Advances in Water Resources in 2015. In that case, I suggested a major revision, but I see that the Authors did not account for most of my suggestions. Some of them are reported below once again with updates.

**Response:**

We thank the referee very much for reconsidering our paper. Concerning the above mentioned incident, we feel really sorry to cause the misunderstanding, and now would like to take this opportunity to clarify this. Indeed, we submitted a paper entitled "Comparing Annual Maximum and Peaks over Threshold methods in nonstationary flood return level analysis" to the journal *Advances in Water Resources* in 2015. This paper was reviewed by two anonymous referees, but due to some unknown reasons we received a decision letter with only one referee's comment in the editor's mail. In that mail, it was just mentioned that the other anonymous referee had suggested a major revision and posted his/her detailed comments as an attachment. However, this attached comment was not found in that mail or in the submission system. We had tried to contact the editor to ask for this missing attachment but without receiving any replies. We have then revised the paper based on the comment of one reviewer and based on our improved understanding of the topic, and decided to submit

the revised paper to *Hydrology and Earth System Sciences*. Once again, we deeply regret for this incident and thank the reviewer for reading our paper two times. We are happy to have this opportunity to respond to reviewer's professional comments.

**General comment 2:**

The main contribution of this study should be the use of the expected-number-of-events (ENE) method and the negative binomial distribution to replace the Poisson distribution under overdispersion conditions. Concerning the ENE method, it is only one of the possible approaches to define return periods and corresponding return levels. It yields the general equation $T = \dfrac{T}{m\sum_{t=1}^{T}(1-F_t(x_T))} = \dfrac{1}{mE[1-F_t(x_T)]}$ , which reduces to the classical $T = \dfrac{1}{m[1-F(x_T)]}$ if $F_t(x_T) \equiv F(x_T)$ for each *t*. In other words, the return period corresponds the expected value of the reciprocal of the exceedance probability of a fixed quantile $x_T$, which is constant if $F_t(x_T)$ is constant. Therefore, sentences such as "This advantage makes the method able to provide unique design value for reference even though the flood behaviors observe nonstationarity, which is beyond the capacity of traditional stationarity strategy" make little sense because return periods and return levels, being expected values taken over *T* (for ENE) or ∞ (for expected waiting time), are always unique values in both stationary and nonstationarity frameworks (a discussion is provided by Serinaldi (2015)).

**Response:**

We thank the referee very much for the valuable and professional remark. We fully agree with the referee's theoretical demonstration for the ENE method. It is really true that the ENE method can provide unique *T*-year return level in both stationary and nonstationary contexts, and on stationary condition, the ENE method will be the same as the traditional stationarity strategy. Actually, in the original paper, we have admitted this important theoretical fact of the ENE method in Lines 315-333.

The sentence "*This advantage makes the method able to provide unique design value … traditional stationarity strategy*" in fact did not refer to the disability of the ENE method when this method is used in a stationary framework, but meant that the traditional stationarity strategy, assuming that the exceedance probability of *T*-year return period equals to $1-F(x_T) = 1/(mT)$ , cannot offer unique design values $x_T$ when nonstationary distribution model with time-varying parameters has been applied. For example, with the use of the LNO3 model in the original manuscript, the 50-year AM flood return level (corresponding to the exceedance probability of 0.02) as shown by the red line in Figure 1 below varies from year to year. The similar examples have been earlier reported in previous literatures (e.g., Villarini

et al., 2009; López and Francés, 2013). In the newly revised manuscript, we have rephrased the relevant sentences to avoid communicating this misleading message.

[Figure]

Figure 1 50-year return levels ($m^3$/s) (corresponding to the exceedance probability of 0.02) estimated with the stationary (black line) and nonstationary (red line) LNO3 models, respectively.

Reference

López, J., and Francés, F.: Non-stationary flood frequency analysis in continental Spanish rivers, using climate and reservoir indices as external covariates, Hydrol. Earth Syst. Sci., 17, 3189-3203, doi:10.5194/hess-17-3189-2013, 2013.

Villarini, G., Smith, J.A., Serinaldi, F., Bales, J., Bates, P.D., and Krajewski, W.F.: Flood frequency analysis for nonstationary annual peak records in an urban drainage basin, Adv. Water Resour, 32, 1255-1266, doi:10.1016/j.advwatres.2009.05.003, 2009.

**General comment 3:**

As far as the negative binomial distribution is concerned, it was already discussed in stationary flood frequency analysis, and compared with Poisson by Bhunya et al. (2013), while its introduction and theoretical justification in stationary and nonstationarity frameworks was presented by Eastoe and Tawn (2010). In particular, the latter highlighted how the overdispersion is not necessarily a consequence of nonstationarity. In fact, overdispersion can easily results from (hidden) persistence (see e.g. Serinaldi F, Kilsby CG. (2016a) and references therein) and/or mixed effects (random fluctuations of the rate of occurrence). Moreover, saying that "over-dispersion of observations" is a possible source of nonstionarity (P9L161- 168) seems to me logically flawed because overdispersion is not a cause but an effect of non-Poissonian behaviour, which can have many different causes. Moreover, other models such as generalized Poisson can be used (Raschke, 2015). Again, nonstationarity is not a necessary condition. However, what really matters is that the distribution of the number of event under nonstationarity is

neither Poisson nor negative binomial, but Poisson binomial (Tejada and den Dekker, 2011; Obeysekera and Salas, 2016). Thus, a more careful literature review should be performed before running (a bit blindly) computer codes/packages.

**Response:**

Great thanks to the referee for the valuable and professional remark and pointing out the inaccurate statement. Sorry that we failed to make clear distinction between the important concepts of nonstationarity and over-dispersion. In the revised manuscript, we have corrected the wrong wording of "two-type source of nonstationarity" to clarify these two concepts.

In this study, we have employed the NB distribution (Anscombe, 1950) as an alternative to the Poisson distribution. There are four main considerations that motivate us to adopt the Negative Binomial (NB) distribution as follows:

(1) The NB distribution is a two-parameter mixture of the Poisson distribution and includes the Poisson distribution as a special case (Please refer to Table 1 for the probability mass function in the original manuscript).

(2) The NB distribution is theoretically justifiable for describing over-dispersed data (variance-to-mean ratio greater than unity), while the assumption of Poisson distribution is invalid as it only allows a fit of equi-dispersed data whose variance and mean is identical.

(3) The arrival rate of POT flood has been reported to be over-dispersed in many available literatures. The NB distribution has received a wide use to accommodate the over-dispersion of POT arrival rate (Ben-Zvi, 1991; Önöz and Bayazit, 2001; Silva et al., 2012; Bhunya et al., 2013). Most studies have applied a stationary NB model (with constant parameters) to fit the over-dispersed data (e.g., Cunnane, 1979; Bhunya et al., 2013), and only a few ones have focused on evaluating whether the over-dispersed data have also shown a nonstationary behavior over a certain long time period. Therefore, this study has been intended to supplement some gaps in studies with the over-dispersed data, in which the accuracy of both stationary and nonstationary (with time-varying parameters) NB models has been evaluated in flood return level analysis.

(4) The requirement of independent data is relaxed when we fit a nonstationary NB model (which is strictly required by the Poisson model, whether with constant or time-varying Poisson process intensity). This advantage has made the NB distribution become increasingly popular especially when it is doubtful whether the observed arrival rates from a stochastic process satisfy the assumption of independence (Johnson et al., 1992).

Within the above background, we believe that the use of NB distribution is in the scope of this study and should be sufficient for the current purpose. It should mention that there have

been different studies of other discrete distributions proposed for describing count data, such as the generalized Poisson (Johnson et al., 1992), the Poisson binomial distribution (Obeysekera and Salas, 2016), or other mixed/derivative distributions (Johnson et al., 1992). Since the adoption of a single type of optimum distribution for POT arrival rate is usually inconclusive in practical application, all the above distributions have been ever successfully used for a specific basin with their respective advantages, e.g., the Poisson binomial distribution, derived as a convolution of Poisson and binomial distributions (Johnson et al., 1992), allows to describe the under-dispersion (i.e., variance is lower than mean). The Poisson binomial distribution is a useful tool that may be applied to model the frequency of POT events (Obeysekera and Salas, 2016) but not the single way to realize this. Although incorporation of other applicable distributions would be lengthy for this paper, it should be a very interesting and meaningful topic for further research.

Following the referee's suggestion, we have carefully reviewed the relevant literatures and summarized them briefly in the newly revised manuscript. For the purpose of giving a better understanding of the NB distribution to the readers, we have added the explanation of theoretical background of the NB distribution, the test of over-dispersion before conducting calculation, and revised the text in the related part of the revised version.

Reference

Anscombe, F.J.: Sampling theory of the negative binomial and logarithmic series distributions, Biometrika, 37, 358-382, doi:10.2307/2332388, 1950.

Ben-Zvi, A.: Observed advantage for negative binomial over Poisson distribution in partial duration series, Stoch. Hydrol. Hydraul., 5, 135-146, doi:10.1007/BF01543055, 1991.

Bhunya, P. K., Berndtsson, R., Jain, S. K., and Kumar, R.: Flood analysis using negative binomial and Generalized Pareto models in partial duration series (PDS), J. Hydrol., 497, 121-132, doi:10.1016/j.jhydrol.2013.05.047, 2013.

Cunnane, C.: A note on the Poisson assumption in partial duration series models, Water Resour. Res., 15, 489-494, doi:10.1029/WR015i002p00489, 1979.

Johnson, N. L., Kemp, A. W., and Kotz, S.: Univariate Discrete Distributions, Second Edition, New York, John Wiley and Sons, 1992.

Önöz, B., and Bayazit, M.: Effect of the occurrence process of the peaks over threshold on the flood estimates, J. Hydrol., 244, 86-96, doi:10.1016/S0022-1694(01)00330-4, 2001.

Obeysekera, J. and Salas, J. D.: Frequency of Recurrent Extremes under Nonstationarity, J. Hydrol. Eng., 21, doi: 10.1061/(ASCE)HE.1943-5584.0001339, 2016.

Silva, A.T., Naghettini, M. and Portela, M.M.: On some aspects of peaks-over-threshold

modeling of floods under nonstationarity using climate covariates, Stoch. Environ. Res. Risk Assess., 1-18, doi:10.1007/s00477-015-1072-y, 2015.

**General comment 4:**

Most of the conclusions in the case study are quite speculative because the behaviour of the return periods under nonstationarity depends on many factors, such as the link functions, the relationships between distributions' parameters and covariates (linear or polynomial regression are surely convenient but also quite arbitrary and surely not physically based), as well as the nature of the distributions (fat tailed, heavy tailed, etc.). In this respect, conclusions are quite fair as they reflect the overall uncertainty of the empirical results, which is however exacerbated by lack of theoretical reasoning on the rationale and true nature of the methods used.

**Response:**

We thank the referee very much for the insightful comment/remark. We fully understand the reviewer's concern about the uncertainty involved in the return levels estimated in a nonstationary context. Admittedly, flood return level analysis comprises many procedures for sampling scheme, probability distribution model, and return level concepts. Each procedure entails many assumptions and would introduce uncertainties (e.g., choice of data, regression modeling, fitting technique; operational decision) (Yen, 2002). Therefore, improvement of reliability of flood return level is still a big challenge in hydrologic studies, not only on nonstationary but also on stationary conditions. In this study, some efforts have actually been made to relieve the impact of uncertainties in nonstationary flood return level analysis as much as possible, such as the effect of the choice of data, importance of preliminary diagnosis and attribution analysis, modeling of probability distribution including the covariates-dependent relationship, and to better understand the overall uncertainty in nonstationary flood return level analysis, we have also performed a sensitivity analysis to study how flood return level would be influenced by changing climate, etc.

In the revised manuscript, we have intended to improve the elaboration on theoretical reasoning and the methods used, and illustrated the limitation and future direction of the nonstationary flood return level analysis.

Reference

Yen, B.C.: System and component uncertainties in water resources. Risk, Reliability, Uncertainty, and Robustness of Water Resources Systems, Bogardi, J.J., Kundzewicz, Z.W. (eds), International Hydrology Series. Cambridge University Press: Cambridge, 133-142,

2002.

**General comment 5:**

By the way, it is worth noting that the models with parameters depending on covariates that exhibit stochastic fluctuations (such as rainfall and temperature) are not nonstationary but simply doubly stochastic. Nonstationary models require that the distribution (marginal and joint) change with time according to some well-defined function holding true for whatever instant along the time axis. In this respect, trend analysis can only detect local changes in a very small interval (i.e., the period of record), and this explains why nonstationarity cannot be inferred from trend analysis but requires exogenous information, i.e. attribution based on physical reasoning rather than statistical correlation analysis. Some additional specific remarks are provided below. I also refer the Authors to my report for the previous AWR version.

**Response:**

We thank the referee very much for this valuable and thought-provoking remark, which has prodded us into thinking more deeply on the concept of nonstationarity. Here, we would like to address this remark from four main aspects beginning with the question about nonstationarity and make a discussion:

(1) *What should be nonstationarity?* For a very long interval on the time axis, the so-called significant (local) changes maybe in fact belong to parts of the behavior in stochastic fluctuations since the observed hydrological records is very short compared with the infinite hydrological process. A typical example can be assumed with a stochastic regime-switching process (Serinaldi and Kilsby, 2015), which has been usually occurred in climatic variable with a periodicity characterized by the alternatively positive and negative variations. Both positive and negative variations are locally persistent fluctuations rather than the behaviors of nonstationarity. Therefore, the observed changes (in a small time interval) do not necessarily imply nonstationarity, and certainly stationarity does not contradict (local) changes (Montanari and Koutsoyiannis, 2014).

(2) *How can nonstationarity be inferred?* It is true that nonstationarity cannot be inferred simply from the result of statistical tests on a finite observation sample. The problem in the understanding of nonstationarity is that the complex natural process that evolves over time can never suffice for the requirement of the representative observation of population. No matter how large the sample size of observation is, the observed series that is updated and prolonged along time axis will always be the segment of a natural process. In order to infer the real nonstationarity, in addition to the observation data alone, detection of

nonstationarity requires exogenous information to explicitly interpret the physical mechanism of nonstationarity and further to guarantee that the patterns observed in a time slice is not just an effect of fluctuations of a stationary process (Serinaldi and Kilsby, 2015).

(3) *How can nonstationarity be described?* We accept the viewpoint that nonstationarity is justifiable if it can be well-defined by a deterministic model that is right for whatever instant along the time axis. Therefore, the current use of a nonstationary model is merely a modelling option on the basis of observation data (Serinaldi and Kilsby, 2015), which might be in fact devised to model the irregular changes in globally stationary process. However, we have to admit one fact that even if a true perfect deterministic function really exists for nonstationary case and holds true along the entire time axis, it can never be exactly known given the complexity and dynamic nature of hydrologic system (Milly et al., 2015), but estimated based on the data and await perpetually the test by time when longer observations become available. This is very different from the case of synthetically stochastic process predefined already by a mathematical construction to generate a large number of samples, as has been exemplified in some literatures (e.g., Koutsoyiannis and Montanari, 2014). Indeed, the nonstationary model may be not the best choice for modeling a nonstationary process as it is unable to ensure a perfectly reliability for future application. A more justified description of nonstationarity based on the theory of stochastic process or dynamic systems has been developed and suggested in literatures (Koutsoyiannis, 2006; Montanari and Koutsoyiannis, 2012, 2014; Koutsoyiannis and Montanari, 2014; Serinaldi and Kilsby, 2015), which would advance the development of our future studies in modeling of the nonstationarity in the hydrological series.

(4) *How did we handle nonstationarity?* In this paper, we would like to talk about nonstationarity in a conservative way, i.e., to use the word nonstationarity to simply mean that the observed time series is no longer identically distributed over a certain time period. This use implies the property of distribution models but not to say that the nature of a natural process is nonstationary, which has in fact been widely acceptable and applied in most of studies that have carried out the investigation of nonstationarity in flood series (Villarini and Serinaldi, 2012; López and Francés, 2013; Salas and Obeysekera, 2014). Here the nonstationarity, as the referee pointed out, is valid only for a finite time period depending on the data record used, and should be a local change (relative to the very long time). The nonstationarity should have actually existed in the hydrological series in many basins worldwide under the changing climate (IPCC, 2013), and in the Weihe basin, China, the significant variability in the natural river regime has been reported repeatedly

before (Zuo et al., 2012; Du et al., 2015), and has been prove to be mainly ascribed to the impact of climate change (Xiong et al., 2014; Jiang et al., 2015).

In the revised manuscript, we have explicitly demarcated the nonstationarity used in this study and made relevant discussion on the concept of nonstationarity in a natural process.

Reference

Du, T., Xiong, L., Xu, C.-Y., Gippel, C.J., Guo, S., and Liu, P.: Return period and risk analysis of nonstationary low-flow series under climate change, J. Hydrol., 527, 234-250, doi:10.1016/j.jhydrol.2015.04.041, 2015.

IPCC 2013 Climate Change 2013: The Physical Science Basis. Contribution of Working Group I to the Fifth Assessment Report of the Intergovernmental Panel on Climate Change. Cambridge University Press, Cambridge, United Kingdom and New York, NY, USA.

Jiang, C., Xiong, L., Wang, D., Liu, P., Guo, S., and Xu, C.-Y.: Separating the impacts of climate change and human activities on runoff using the Budyko-type equations with time-varying parameters, J. Hydrol., 522, 326-338, doi:10.1016/j.jhydrol.2014.12.060, 2015.

Koutsoyiannis, D.: Nonstationarity versus scaling in hydrology, J. Hydrol., 324, 239-254, doi:10.1016/j.jhydrol.2005.09.022, 2006.

Koutsoyiannis, D.: Hurst-Kolmogorov dynamics and uncertainty, J. American Water Resour. Association, 473, 481-495, doi:10.1111/j.1752-1688.2011.00543.x, 2011.

Koutsoyiannis, D., and Montanari, A.: Negligent killing of scientific concepts: The stationarity case, Hydrol. Sci. J., 60, 1174-1183. doi:10.1080/02626667.2014.959959, 2014.

Milly, P.C.D., Betancourt, J., Falkenmark, M., Hirsch, R.M., Kundzewicz, Z.W., Lettenmaier, D.P., Stouffer, R.J., Dettinger, M.D., and Krysanova, V.: On Critiques of "Stationarity is Dead: Whither Water Management?", Water Resour. Res., 51, 7785-7789, doi:10.1002/2015WR017408, 2015.

Montanari, A., and Koutsoyiannis, D.: A blueprint for process-based modeling of uncertain hydrological systems, Water Resour. Res., 48, W09555, doi:10.1029/2011WR011412, 2012.

Montanari, A., and Koutsoyiannis, D.: Modeling and mitigating natural hazards: Stationarity is immortal!, Water Resour. Res., 50, 9748-9756, doi:10.1002/ 2014WR016092, 2014.

Salas, J. D., and Obeysekera, J.: Revisiting the Concepts of Return Period and Risk for Nonstationary Hydrologic Extreme Events, J. Hydrol. Eng., 19(3), 554-568,

doi:10.1061/(ASCE)HE.1943-5584.0000820, 2014.

Serinaldi, F., and Kilsby, C. G.: Stationarity is undead: Uncertainty dominates the distribution of extremes, Adv. Water Resour., 77, 17-36, doi:10.1016/j.advwatres.2014.12.013, 2015.

Villarini, G., and Serinaldi, F.: Development of statistical models for at-site probabilistic seasonal rainfall forecast, Int. J. Climatol. 32, 2197-2212, doi:10.1002/joc.3393, 2012.

Xiong, L., Jiang, C., and Du, T.: Statistical attribution analysis of the nonstationarity of the annual runoff series of the Weihe River, Water Sci. Technol., 70, 939-946, doi:10.2166/wst.2014.322, 2014.

Zuo, D., Xu, Z., Yang, H., and Liu, X.: Spatiotemporal variations and abrupt changes of potential evapotranspiration and its sensitivity to key meteorological variables in the Wei River basin, China, Hydrol. Process., 26, 1149-1160, doi:10.1002/hyp.8206, 2012.

**Specific comments**

(1) L141-143: see Serinaldi (2015) for a wider discussion.

**Response:**

Thanks very much for this comment. We have carefully followed this reference and supplemented the relevant contents in the revised manuscript.

(2) L146: "other sampling methods. . .seem"

**Response:**

Thanks, it has been revised as suggested.

(3) L189: TFPW does not perform any prewhitening and does not preserve the nominal significance level. This explains why the results reported in the literature for MK and TFPW MK are often close to each other (see Serinaldi and Kilsby (2016b) for an analytical and numerical proof)

**Response:**

We thank the referee very much for the professional comment. We have carefully followed the reference provided by the referee and learned more from it. In the newly revised manuscript, we have corrected the use of original TFPW method to make real pre-whitening and pointed to this reference for the relevant theoretical basis.

(4) L202-205: The interpretation of the partial MK test is not correct. Moreover, the interpretation reflects a widespread merging of Neyman-Pearson testing procedure and Fisher's p-values, whose values cannot be interpreted as proxies of the strength of the

relationship between target variables and covariates.

**Response:**

We thank the referee very much for pointing out the improper expression. The misleading statement has been deleted in the newly revised manuscript.

(5) L268-275: AIC and BIC have different meaning and are relative measures. Thus, model selection should be based on Akaike weights and/or evidence ratios (see Burnham and Anderson (2002, 2004).

**Response:**

Thanks a lot for this suggestion. In the newly revised manuscript, we have applied the Akaike weights for model selection as suggested. However, we have retained AIC considering that it has received a wide use in hydrologic studies (BIC has been deleted as it is very similar to AIC).

(6) L289: If the process is not stationary, the empirical frequencies cannot be computed by the Gringorten formula. Classical qq plot does not make sense in a (true) nonstationary framework. Moreover, in GAMLSS, the residuals are not the normal quantile transform of the observed values (this holds only for stationary models) but the difference between the predictions (given the covariates) on the observations (for the same covariates). Furthermore, qq plots and coefficient of determination are not formal tests but diagnostic plots and measures of performance, respectively. In particular, no tests can be performed at the 5% significance level for coefficients of determination (unless ad hoc MC experiments are set up).

**Response:**

We thank the referee very much for this professional comment. It is true that the Gringorten formula should not be used to calculate the empirical frequency of the observations that have been assumed to be described by nonstationary distribution model with time-varying parameters, and the classical qq plot would be meaningless if it is routinely made for these observations.

Actually, in the original manuscript, we have introduced the residuals $r_t$ of nonstationary distribution model (with time-varying parameters), which can be applied to the Gringorten formula and the classical qq plot. The calculation of the residuals $r_t$ follows two steps, i.e., by inverting the well-established nonstationary distribution model into the cumulative probabilities at each time $t$ and then transforming these cumulative probabilities to the standard normalized quantiles (Dunn and Smyth, 1996). For example, denote $x_t$ being

observation value at time $t$ that is fitted by a nonstationary GEV model with time-varying parameters $\theta_t$. $F_{\mathrm{GEV}}(\cdot)$ represents the cumulative distribution function. The cumulative probabilities at time $t$ should be $F_{\mathrm{GEV}}(x_t|\theta_t)$. The residual $r_t$ of the fitted GEV model is calculated as $r_t = \Phi^{-1}(F_{\mathrm{GEV}}(x_t|\theta_t))$, where $\Phi^{-1}$ is the inverse function of standard normal distribution. It follows from its definition that the residuals $r_t$ are exactly standard normal when $\theta_t$ is constant (i.e., for the stationary GEV model), and when nonstationary model fits data well, the residuals $r_t$ should converge to standard normal distribution (Dunn and Smyth, 1996). Therefore, the empirical frequency formula is applicable for $r_t$. The classical qq plot can be used to test the normality of $r_t$ for evaluating how well the nonstationary model (with time-varying parameters) fits data.

In the newly revised manuscript, we have revised the relevant text to elaborate clearly the residuals $r_t$. Following the referee's suggestion, the qq plots and coefficient of determination have been deleted as they cannot give a quantifiable result of statistical tests. Instead, we have employed the worm plots and Filliben correlation coefficient (Filliben, 1975) for goodness-of-fit tests, which have been widely used for evaluating the performance of model at the 5% significance level (Villarini et al., 2010; López and Francés, 2013).

Reference

Dunn, P.K., and Smyth, G.K.: Randomized quantile residuals, J. Comput. Graph. Stat., 5, 236-244, doi:10.2307/1390802, 1996.

Filliben, J.J.: The probability plot correlation coefficient test for normality, Technometrics, 17: 111-117, doi:10.1080/00401706.1975.10489279, 1975.

López, J., and Francés, F.: Non-stationary flood frequency analysis in continental Spanish rivers, using climate and reservoir indices as external covariates, Hydrol. Earth Syst. Sci., 17, 3189-3203, doi:10.5194/hess-17-3189-2013, 2013.

Villarini, G., Smith, J. A., and Napolitano, F.: Nonstationary modeling of a long record of rainfall and temperature over Rome, Adv, Water Resour., 33, doi:10.1016/j.advwatres.2010.03.013, 1256-1267, 2010.

(7) L494-496: This result is not so surprising because the number of exceedances decreases as the threshold increases, and therefore the clustering of extreme events is more evident given the short time series.

**Response:**

Thanks a lot for this valuable comment. We have reframed the sentence according to this

remark in the newly revised manuscript.

(8) Section 4.3: GAMLSS are nothing but an advanced form of regression. Using the fitted model with covariates taking values beyond the fitting range is never a good idea because we do not know if the fitted relationship still holds true in that range.

**Response:**

We thank the referee for this valuable comment. We fully agree that the fitted best model with parameters as functions of climatic covariates may not hold true for future prediction as no one can tell what the future really should be like. Actually, the extrapolation of future design floods by using the best fitted nonstationary model pre-determined with historical data has an underlying assumption that the priori model used for projection is acceptably reasonable at present and in the near future (Du et al., 2015; Milly et al., 2015). This assumption is built on the consideration that flood processes themselves may present correlation or locally persistent fluctuations over a certain time period. For the practical point of view, we would like to say it is not the best idea but a realistically acceptable and useful idea, which serves the practical need for prediction in design, planning, and management over the decades-long design horizon of engineering (e.g., related to a specific multi-year or short-term project plan for a certain future design period). Ignoring the detected local change induced by climate change for making prediction over a certain design period may cause large estimation errors if only considering a single use of stationarity strategy. Therefore, the current nonstationary flood return level analysis is aimed at what should be done when nonstationarity (or short-term local changes) really happen and how to understand the uncertainty when this nonstationarity has been considered in modeling and predicting floods.

Within the above background, this study is intended to make a contribution for practical applications in a nonstationary design framework over a certain time period and provide an alternative choice for decision-makers when significant changes have been informed and destroyed the identically distributed assumption in engineering design. The present study has attempted to seek a way to achieve multi-decadal flood projections, which enables us to obtain a short-term foresight of the variation tendency of flood return levels, and performed a global sensitivity analysis to help understand how the design floods would be influenced by the changing climate or how large the possible overall uncertainty would be if the pre-determined nonstationary model is applied to future application. It should mention that the current method of nonstationary flood return level analysis is in fact devised based on the observed local changes but not the true understanding of global variations of natural process, and thereby admittedly far from perfect. The researchers should always consider the

stationary flood frequency analysis, an effective tool to handle the possibility of irregular local variations in a stationary process and avoid a misuse of nonstationarity for the very long future.

To address this comment, in the revised manuscript, we have illustrated more explicitly the assumption that the extrapolation analysis of future flood with nonstationary models should not be readily given for a very distant future but confined in a specific time period during which the pre-defined nonstationary model can be practically acceptable.

Reference

Du, T., Xiong, L., Xu, C.-Y., Gippel, C.J., Guo, S., and Liu, P.: Return period and risk analysis of nonstationary low-flow series under climate change, J. Hydrol., 527, 234-250, doi:10.1016/j.jhydrol.2015.04.041, 2015.

Milly, P.C.D., Betancourt, J., Falkenmark, M., Hirsch, R.M., Kundzewicz, Z.W., Lettenmaier, D.P., Stouffer, R.J., Dettinger, M.D., and Krysanova, V.: On Critiques of "Stationarity is Dead: Whither Water Management?", Water Resour. Res., 51, 7785-7789, doi:10.1002/2015WR017408, 2015.

(9) English should be revised and some typos fixed.

**Response:**

Thanks, we would like to employ an English editing service to improve the writing quality of the newly revised manuscript.

Once again, many thanks for your professional and valuable comments which greatly improve our research and paper.

With best wishes

Yours sincerely

Lihua Xiong, PhD, Professor

State Key Laboratory of Water Resources and Hydropower Engineering Science

Wuhan University

Wuhan 430072, PR China

E-mail: xionglh@whu.edu.cn

Telephone: +86-13871078660

Fax: +86-27-68773568

---

## Author Comment (AC3) · 2 Mar 2017

**Responses to Referee #3**

**Dear Referee #3,**

We are very grateful to you for the professional and constructive comments on our manuscript entitled "Comparative study of flood projections under the climate scenarios: links with sampling schemes, probability distribution models, and return level concepts" (Number: hess-2016-619). These comments are all valuable and very helpful not only for improving this paper but also beneficial for our research. We have carefully followed these comments and accordingly made the relevant revisions. The point-by-point responses to the comments and the corresponding correction to the manuscript are explained as follows:

**General comment 1:**

In the submitted paper authors compare different methods that can be selected to perform the flood frequency analysis (FFA). Both stationary annual maximum (AM) method and peaks over threshold (POT) method are tested using the daily discharge data from the Huaxian station (Weihe basin) in China. Further, a nonstationary methodology is also applied and compared with the stationary approach. Several interesting and important aspects of the flood frequency analysis approach are analyzed and discussed: application of the expected-number-of-events (ENE) method, selection of different distribution functions in the AM method, evaluation of the suitability of the Poisson and negative binomial (NB) distributions to model the annual number of exceedances above the threshold, comparison between the stationary and nonstationary approaches using both AM and POT methods, sensitivity analysis regarding the influence of the precipitation and temperature on the nonstationary flood frequency analysis results. The paper is relatively well written and the presented topic is interesting for the hydrological society due to the importance of the flood frequency approach for the design of different hydro-technical structures. The paper is in the scope of the journal. However, I would suggest that authors try to put more focus on next points related to the practical applications of the FFA, because the presented paper does not develop a new theory but compares different aspects of the FFA approach:

**Response:**

We thank the referee very much for the positive evaluation concerning our manuscript. The following comments have been addressed point by point as below.

**General comment 2:**

2.1 The authors have performed detailed analysis and comparison of different methods that can be used to carry out the flood frequency analysis. Is it possible to point out which method should be used by practitioners to determine the design flood (taking into account the larger sensitivity of the nonstationary AM method compared to the POT, more complicated POT analysis compared to the AM method and other conclusions stressed in this paper)

**Response:**

Great thanks to the referee for this constructive comment and suggestion. We would like to address this comment from three aspects in accordance with the title of this paper:

(1) *Concerning the sampling schemes.* As we stated in the original manuscript, we suggest that the POT sampling scheme should receive more attention (or at least as much as the AM) in flood return level (RL) analysis, especially when research is conducted on nonstationary condition. First, the POT can describe not only the magnitude of extreme events but also the frequency of the events. Changes in the frequency of extreme events have increasingly attracted public attention (Villarini et al., 2012), and the POT is an effective tool to study this. Second, based on the results of this study, RL derived with POT shows a lower sensitivity to climate change than that with AM. This implies that the established nonstationary POT model, when extrapolated to future application, may have less uncertainty introduced by the climate scenarios than the AM. The plausible reason has been given in Specific comment (22). Third, the sampling method of AM, which stipulates one event per year, is simple to perform, but may lose the significance of real flood and thereby leads to uncertainty of RL. Although the POT sampling method has the uncertainty problem brought by the complicated sampling criteria (e.g., threshold selection), this problem may be relieved to some extent by selecting a range of acceptable thresholds and then comparing the RL results of them. Finally, there is often the case of short data records that renders the inapplicability of AM for both statistical analysis and operational application (due to the insufficient representativeness of population). However, the POT sampling scheme can offer longer sample size and more than one time series. It is found from Figure 5 in the original manuscript that on both stationary and nonstationary conditions, RL derived with POT series yield narrower confidence limits than the AM series.

(2) *Concerning the probability distribution models.* According to the specific study on description of POT arrival rate, the negative binomial (NB) model has been demonstrated to be superior to the Poisson model when POT arrival rate shows a behavior of

over-dispersion. This result coincides with the theoretical basis of the NB distribution that is characterized by variance-to-mean ratio greater than unity. Besides, the NB distribution includes the Poisson distribution as a special case (Anscombe, 1950). When it is doubtful whether the observed POT arrival rate really comes from Poisson distribution, the NB distribution can be preferable for practical application. For both modeling the nonstationarity and making prediction, covariates with physical meaning have been proven to be more reasonable than those without a cause-effect relationship with the nonstationarity of extreme events (e.g., time) (López and Francés, 2013; Du et al., 2015; Prosdocimi et al. 2015). In the real-world application, practitioners have to carefully select the appropriate physical covariates that should be closely related to the flood variability in the actual study region.

(3) *Concerning the return level concept.* In hydrologic studies, flood return level analysis has been conducted with either the assumption of traditional stationarity strategy or the methods adapted to nonstationarity. From the perspective of the practitioners, it is more advisable to employ the method developed with nonstationarity if flood variability really happens over a certain long period, together with a sensitivity/uncertainty analysis as the current work did to enhance the understanding of the possible impact of nonstationarity on return level estimation. But above all, whenever nonstationarity has been considered for flood extrapolation, traditional stationarity strategy must be always retained as the baseline inference. Please see more explanations in the response to General comment 2.2. The ENE method studied in this paper is one of the pertinent ways to extrapolate design floods for both AM and POT series on either stationary or nonstationary condition. Further studies can focus on more other (e.g., expected waiting time) or new methods, especially for POT series, to carry out broader studies on future flood projection.

According to the referee's suggestion, in the revised manuscript, we have added a brief discussion on the practical application of nonstationary flood return level analysis.

Reference

Anscombe, F.J.: Sampling theory of the negative binomial and logarithmic series distributions, Biometrika, 37, 358-382, doi:10.2307/2332388, 1950.

Du, T., Xiong, L., Xu, C.-Y., Gippel, C.J., Guo, S., and Liu, P.: Return period and risk analysis of nonstationary low-flow series under climate change, J. Hydrol., 527, 234-250, doi:10.1016/j.jhydrol.2015.04.041, 2015.

López, J., and Francés, F.: Non-stationary flood frequency analysis in continental Spanish rivers, using climate and reservoir indices as external covariates, Hydrol. Earth Syst. Sci., 17,

3189-3203, doi:10.5194/hess-17-3189-2013, 2013.

Milly, P.C.D., Betancourt, J., Falkenmark, M., Hirsch, R.M., Kundzewicz, Z.W., Lettenmaier, D.P., Stouffer, R.J., Dettinger, M.D., and Krysanova, V.: On Critiques of "Stationarity is Dead: Whither Water Management?", Water Resour. Res., 51, 7785-7789, doi:10.1002/2015WR017408, 2015.

Prosdocimi, I., Kjeldsen, T.R., and Miller, J.D.: Detection and attribution of urbanization effect on flood extremes using nonstationary flood frequency models, Water Resour. Res., 51, 4244-4262, doi:10.1002/2015WR017065, 2015.

Villarini, G., Smith, J.A., Serinaldi, F., Ntelekos, A.A., and Schwarz, U.: Analyses of extreme flooding in Austria over the period 1951-2006, Int. J. Climatol., 32, 1178-1192, doi:10.1002/joc.2331, 2012.

**2.2 Should the standard procedures to perform the FFA in China be modified after the results of this study?**

**Response:**

Thank you very much for this constructive and valuable comment. For practical application, the more realistic suggestion is to take into account nonstationarity when flood observations really exhibit significant changes at a certain long time period, carefully analyze the physical mechanism behind the detected changes in floods to select reasonable explanatory (physical) covariates, and extrapolate design floods to the near future together with a sensitivity/uncertainty analysis to enhance the understanding of possible impact of physical covariates on nonstationary flood return level analysis. In addition, as we stated in the introduction and actually did in this study, the standard procedures of stationarity strategy to perform the flood return level analysis should always be retained as the baseline for references. This solution can anyway be more informative than the insistence of single use of the traditional standard to perform flood frequency analysis while variability in flood really happens.

It is worth mentioning that the above suggestion should be practicably acceptable and very useful only over the decades-long design horizon of engineering (e.g., some projects are always related to a specific multi-year design plan) but not the very distant future. On the one hand, the presence of flood variability caused by the changing climate cannot readily be neglected as the variability in flood series (that leads to not identically distributed sample) may cause big uncertainty for both modeling and estimating floods when flood return level analysis is still performed by traditional stationarity strategy (Milly et al. 2015). On the other hand, due to incorporation of nonstationarity, model complexity certainly will increase, and

thereby could also induce other sources of uncertainty. Therefore, the extrapolation of design floods should be confined in a specific scope with the underlying assumption that the pre-determined nonstationary distribution model for flood extrapolation will be applicable for a near future.

Reference:

Milly, P.C.D., Betancourt, J., Falkenmark, M., Hirsch, R.M., Kundzewicz, Z.W., Lettenmaier, D.P., Stouffer, R.J., Dettinger, M.D., and Krysanova, V.: On Critiques of "Stationarity is Dead: Whither Water Management?", Water Resour. Res., 51, 7785-7789, doi:10.1002/2015WR017408, 2015.

**2.3 What is the trade-off (if any) between the model complexity and uncertainty in the flood frequency analysis results?**

**Response:**

Thanks for this constructive comment. In this study, there are two main aspects related to the trade-off between the model complexity and uncertainty. The first aspect is to perform flood return level analysis under both stationarity and nonstationarity as we have explained in the response to General comments 2.1 and 2.2. The second aspect is to assume a linear dependence on physical covariates for modeling distribution parameter. This assumption should be regarded as a desirable tradeoff between the high-order modeling of covariates (to reduce bias of fit) and the suspicion of over-fitting (to reduce uncertainty) given the previous referential experience (e.g., Villarini et al., 2009; Salas and Obeysekera, 2014; Xiong et al., 2015a, b). With the progress in understanding and modeling of nonstationarity, newly well-defined mathematical treatment that can better describe the physical relationship with the flood variability should be aspired in future study.

Reference

Salas, J.D., and Obeysekera, J.: Revisiting the concepts of return period and risk for nonstationary hydrologic extreme events, J. Hydrol. Eng., 19, 554-568, doi:10.1061/(ASCE)HE.1943-5584.0000820, 2014.

Villarini, G., Smith, J.A., Serinaldi, F., Bales, J., Bates, P.D., and Krajewski, W.F.: Flood frequency analysis for nonstationary annual peak records in an urban drainage basin, Adv. Water Resour, 32, 1255-1266, doi:10.1016/j.advwatres.2009.05.003, 2009.

Xiong, L., Du, T., Xu, C.-Y., Guo, S., Jiang, C., and Gippel, C.J.: Non-stationary annual maximum flood frequency analysis using the norming constants method to consider

non-stationarity in the annual daily flow series, Water Resour. Manag., 29, 3615-3633, doi:10.1007/s11269-015-1019-6, 2015a.

Xiong, L., Jiang, C., Xu, C.-Y., Yu, K.-X., and Guo, S.: A framework of change-point detection for multivariate hydrological series, Water Resour. Res., 51, 8198-8217, doi:10.1002/2015WR017677, 2015b.

**General comment 3:**

Related to the previous point, different methods yielded diverse FFA results. For example, the 50-year flood was estimated to be between approximately 4000 and 8000 $m^3/s$ with the consideration of the confidence intervals (Fig. 5). Can the authors suggest some guidance for selection of the most appropriate method to carry of the FFA?

**Response:**

We thank the referee very much for this comment. The confidence intervals in the estimations of return levels reflect a certain level of uncertainty involved in the flood inference. Overall, in this study, POT series have much narrower confidence intervals than the AM series, implying that it may provide more practically acceptable design flood values than the AM series when nonstationarity has been taken into account in flood return level analysis. Please refer to the response to General comment 2.1 for more explanations, which is very similar to this comment.

**General comment 4:**

Looking at the results of the nonstationary approach (AM method) shown in Fig. 5 it seems that the return level increases to about 30-year return period and then it is almost constant for larger return periods? Does this means that the 50-year flood is the same as the e.g. 200-year flood? Please explain.

**Response:**

We thank the referee's valuable comment. As questioned by the referee, the result of $T$-year ($30 < T < 90$) return level (RL) derived with AM series on nonstationary condition in Fig. 5, indeed does not increase much as $T$ becomes larger. However, $T$-year RL, e.g., 200-year RL ($x_{T=200}^{non-s}$), should not be taken for granted as a natural extension of this result. We give a brief interpretation with an example of the nonstationary GEV model (with parameters as functions of climatic covariates) and apply this model to derive 50- and 80-year RLs (denote as the values $x_{T=50}^{non-s}$ and $x_{T=80}^{non-s}$ hereinafter) by the ENE method.

Denote $F(\cdot|\boldsymbol{\theta}_t)$ being the GEV distribution function with time-varying parameters $\boldsymbol{\theta}_t$. According to the ENE method, $x_{T=50}^{non-s}$ and $x_{T=80}^{non-s}$ should satisfy the inference formula as

$\sum\limits_{t=t_0+1}^{t_0+50}[1-F(x_{T=50}^{non-s}|\boldsymbol{\theta}_t)]=1$ and $\sum\limits_{t=t_0+1}^{t_0+80}[1-F(x_{T=80}^{non-s}|\boldsymbol{\theta}_t)]=1$ , respectively. The difference in cumulative

probability between $\sum\limits_{t=t_0+1}^{t_0+50}[1-F(x_{T=50}^{non-s}|\boldsymbol{\theta}_t)]=1$ and $\sum\limits_{t=t_0+1}^{t_0+80}[1-F(x_{T=50}^{non-s}|\boldsymbol{\theta}_t)]$ over the level $x_{T=50}^{non-s}$

denotes as $\sum\limits_{t=t_0+51}^{t_0+80}[1-F(x_{T=50}^{non-s}|\boldsymbol{\theta}_t)]$ . If $\sum\limits_{t=t_0+51}^{t_0+80}[1-F(x_{T=50}^{non-s}|\boldsymbol{\theta}_t)]$ is approximate to zero, we have

$\sum\limits_{t=t_0+1}^{t_0+80}[1-F(x_{T=50}^{non-s}|\boldsymbol{\theta}_t)]\approx\sum\limits_{t=t_0+1}^{t_0+50}[1-F(x_{T=50}^{non-s}|\boldsymbol{\theta}_t)]=\sum\limits_{t=t_0+1}^{t_0+80}[1-F(x_{T=80}^{non-s}|\boldsymbol{\theta}_t)]=1$ and thus $x_{T=50}^{non-s}\approx x_{T=80}^{non-s}$ , otherwise,

we have $\sum\limits_{t=t_0+1}^{t_0+80}[1-F(x_{T=50}^{non-s}|\boldsymbol{\theta}_t)]>\sum\limits_{t=t_0+1}^{t_0+50}[1-F(x_{T=50}^{non-s}|\boldsymbol{\theta}_t)]=\sum\limits_{t=t_0+1}^{t_0+80}[1-F(x_{T=80}^{non-s}|\boldsymbol{\theta}_t)]=1$ and thus $x_{T=80}^{non-s}>x_{T=50}^{non-s}$ .

In accordance with these illustrations, the result in this study has indeed found that

$\sum\limits_{t=t_0+51}^{t_0+80}[1-F(x_{T=50}^{non-s}|\boldsymbol{\theta}_t)]$ is approximately zero, and $x_{T=50}^{non-s}\approx x_{T=80}^{non-s}$ (the differences are within

80m$^3$/s).

The aforementioned result is caused by the negative trend found in the AM series and the sensitivity to climate change. Therefore, in this study, under the impact of future changing climate, $T$-year RL ($T>90$ years) derived on nonstationary condition should be determined strictly by the ENE inference at any time. Whether $x_{T=200}^{non-s}$ can be same as $x_{T=50}^{non-s}$ remains to be tested. However, the priori information of climatic scenario required to solve the equation $\sum\limits_{t=t_0+1}^{t_0+200}[1-F(x_{T=200}^{non-s}|\boldsymbol{\theta}_t)]=1$ is insufficient, $x_{T=200}^{non-s}$ could be thus inconclusive. Additionally, according to the response to General comment 2.2, future projection of $x_{T=200}^{non-s}$ might have too big uncertainties, and should be an excessive extrapolation for future.

**General comment 5:**

It would be interesting to make a comparison of the nonstationary approach where the model parameters change with time (e.g., Obeysekera and Salas, 2014; Salas and Obeysekera, 2014; Sraj et al., 2016; Vogel et al., 2011) and not only with P and T. The English is understandable, but it could benefit from some improvements, therefore I recommend editing for English language.

**Response:**

Thanks a lot for this good comment and suggestion. We agree with the referee that it can be more informative to compare the results of nonstationary flood return level analysis when taking time and climatic factors as covariates, respectively. Admittedly, using time rather than climatic covariates has the advantage of simplicity in model structure and ease of extrapolation to future (i.e., the priori information of time as covariate is already known). It is in fact a more popular method and has long been used by hydrologists worldwide. However, as we illustrated in the response to General comment 2.1, many recent studies concerning the

nonstationary analysis of extreme events have concluded that physical covariates-dependent models should be practically more reasonable than those purely employing time as covariate in both description and extrapolation of nonstationarity for application. Actually, the superiority of physical covariate has also been found in this study, for which we would like to show an example here (only the AIC and BIC values for brief) in Table 1 as below.

Table 1 The optimal GEV models using time and physical covariates as covariates, respectively.

| GEV model | Estimated parametric functions | AIC | BIC |
|---|---|---|---|
| Using time $t$ | $\mu_t = 1814.942 - 9.857t$
 $\ln(\sigma) = 6.644$
 $\xi = 0.313$ | 854.8 | 862.5 |
| Using physical covariates $T_{mean}, P_{total}$ | $\mu_t = 1789.594 + 3.818 P_{total} - 215.657 T_{mean}$
 $\ln(\sigma_t) = 9.736 - 0.336 T_{mean}$
 $\xi = 0.108$ | 832.3 | 843.8 |

Inclusion of the results with time as covariate would be lengthy for this paper and seems not very consistent with our study objectives. For example, we analyzed how climate change could influence flood projections by the sensitivity analysis and assuming an increment of climatic covariates. These analyses would be meaningless when using time as covariate which is undoubtedly a monotonically increasing variable.

Therefore, in the revised manuscript, we have directed the interested readers to the studies as mentioned by the referee, in which, time has been used as covariate, instead of a detailed illustration in this study. Following the referee's suggestion, the English writing of the revised manuscript will be improved by a language editing service.

**Specific comments**

(1) Page 13, line 257: I would suggest adding a reference for the GAMLSS package.

**Response:**

Thanks for this suggestion. We have added the reference (Rigby and Stasinopoulos, 2005) in the revised manuscript.

Reference

Rigby, R.A., and Stasinopoulos, D.M.: Generalized Additive Models for Location, Scale and Shape, J. Roy. Stat. Soc., 54, 507-554, doi:10.1111/j.1467-9876.2005.00510.x, 2005.

(2) Page 16, lines 313-314: I would suggest rephrasing this sentence.

**Response:**

Thanks, we have rephrased the sentence in the revised manuscript.

(3) Page 16, line 321: What is "dramatic" or "pointless" for the authors? This can be very subjective, thus I would suggest avoiding such statements.

**Response:**

Thanks a lot for pointing out the improper and subjective statement. In the original paper, the word ("dramatic" or "pointless") is used to imply that $T$-year return level (RL) would have a large range of estimation values if stationarity strategy continues to be employed on nonstationary condition. For example, the 50-year RL estimated by the nonstationary LNO3 model for AM series (corresponding to the exceedance probability of 0.02) varies from year to year with large discrepancy (ranging roughly from 3000 to 9500) as shown by the red line in Figure 1 below (the calculation is based on the data and model used in the original manuscript). This result highlights the improper application of stationarity strategy under nonstationary condition. Following the referee's suggestion, we have rephrased the sentences in the revision of manuscript to correct the improper and subjective statement.

[Figure]

Figure 1 50-year RLs (corresponding to the exceedance probability of 0.02) estimated based on the stationary LNO3 (black line) and nonstationary LNO3 models (red line), respectively.

(4) Page 19, line 387: Which Sensitivity package (a reference should be added)?

**Response:**

Thanks for this comment and suggestion. The reference (Pujol et al., 2017) has been added in the revision.

Reference

Pujol, G. et al.: Package "sensitivity", Version 1.14.0, 2017, (retrieved from https://cran.r-project.org).

(5) Page 21, line 408: Replace "134,800" with "134 800" (and also in some other parts of the manuscript).

**Response:**

Thanks, it has been replaced by "134 800" together with the alterations for other parts in the revised version of manuscript (e.g., 106 498).

(6) Page 21, line 414: Upstream and not downstream?

**Response:**

Thanks for pointing out this mistake. It has been corrected in the revision.

(7) Page 22, line 422: Replace "Thiessen polygon" with "Thiessen polygons".

**Response:**

Thanks, the words "Thiessen polygon" has been replaced by "Thiessen polygons".

(8) Page 24, lines 487-488: Any particular physical reason for this negative trend? It would be interesting to see the discharge data used in study.

**Response:**

Thanks for this comment. The negative trend detected in AM flood series is mainly dominated by the changes in climate as has been specifically studied in the previous literatures (Xiong et al., 2014, 2015; Jiang et al., 2015). In the revised manuscript, we have modified the relevant text to explicitly show the physical reason for this negative trend. Following the reviewer's suggestion, we have added the figures to visually display the variations of the discharge series used in this study.

Reference

Jiang, C., Xiong, L., Wang, D., Liu, P., Guo, S., and Xu, C.-Y.: Separating the impacts of climate change and human activities on runoff using the Budyko-type equations with time-varying parameters, J. Hydrol., 522, 326-338, doi:10.1016/j.jhydrol.2014.12.060, 2015.

Xiong, L., Jiang, C., and Du, T.: Statistical attribution analysis of the nonstationarity of the annual runoff series of the Weihe River, Water Sci. Technol., 70, 939-946, doi:10.2166/wst.2014.322, 2014.

Xiong, L., Du, T., Xu, C.-Y., Guo, S., Jiang, C., and Gippel, C.J.: Non-stationary annual maximum flood frequency analysis using the norming constants method to consider

non-stationarity in the annual daily flow series, Water Resour. Manag., 29, 3615-3633, doi:10.1007/s11269-015-1019-6, 2015.

(9) Page 24, line 492: Again, what does "dramatically" means?

**Response:**

Thanks for pointing out the ambiguous expression. In the revised manuscript, "dramatically" has been replaced by "significantly".

(10) Page 25, lines 499-503: Is this the case for all 22 analyzed stations?

**Response:**

Thanks for this comments. The trend test is in fact conducted on the five series of climatic covariates, i.e., annual total precipitation, annual maximum precipitation on consecutive one, three, and seven days, and annual mean air temperature. The at-site daily total precipitation and daily mean temperature series of the 22 stations have been processed into the areal average series by the method of Thiessen polygons. Then the areal average series provide the series for the selected five climatic covariates. The relevant details have been shown in Lines 409-423 in the original manuscript.

(11) Page 26, lines 526-529: I would suggest rephrasing this sentence.

**Response:**

Thanks for this suggestion. We have reworded this sentence in the revised version.

(12) Page 26, line 537: "much lower" this is subjective; I would suggest using the % to show the difference.

**Response:**

Thanks for this comment. In the revised version, we have revised this vague expression as suggested.

(13) Page 29, lines 569-570: What is reason for this large difference and what does this mean from the perspective of the practitioners?

**Response:**

Thanks a lot for this valuable comment. In the question as raised by the referee, the return level is derived with the GEV model under stationarity ( $x_T^s$ ) and nonstationarity ( $x_T^{non-s}$ ) for the future period. The difference between $x_T^s$ and $x_T^{non-s}$ should be attributed to the significantly decreasing trend in the AM series, the significant increase in future temperature and the

higher sensitivity of $x_T^{non-s}$ to temperature. Assuming that the return level derived under nonstationarity remains invariant as $x_T^s$ and substituting $x_T^s$ into the nonstationary GEV distribution function $F(\cdot|\boldsymbol{\theta}_t)$, the exceedance probability of each year $1-F(x_T^s|\boldsymbol{\theta}_t)$ will become lower and lower as $t$ increases, the obtained cumulative exceedance probability of $T$ years satisfies $\sum_{t=t_0+1}^{t_0+T}[1-F(x_T^s|\boldsymbol{\theta}_t)]<1$, which implies that the return period is no longer $T$ years under nonstationarity. To satisfy the ENE inference formula $\sum_{t=t_0+1}^{t_0+T}[1-F(x_T|\boldsymbol{\theta}_t)]=1$, we can thus obtain $x_T^{non-s}<x_T^s$. The result indicates the uncertainty in $x_T^s$ as introduced by the climate change when flood return level analysis is performed without consideration of the impact of nonstationarity while it has existed in flood series.

(14) Page 29, lines 583-585: These are relatively large differences. Which POT threshold is suggested by the authors and why?

**Response:**

Thanks for this valuable comment. The large differences among the results of POT2, POT3, and POT4 mainly originate from the different threshold values $u$. In the newly revised manuscript, we have added the test of mean excess plot for the selected threshold of POT2, POT3, and POT4. The results indicate that the POT4 threshold almost approaches (but is within) the lower bound of domain where the mean excess should be an approximately linear function of $u$ for a valid choice of the generalized Pareto (GP) distribution. Therefore, POT4 series might not be the best appropriate choice in this study. There is no evidence of invalidation of threshold test for POT2 and POT3, both of which should be thus taken as more reasonable choices for practical application. From the perspective of engineering security, POT2 with higher threshold value might be preferred as it ensures a high level of flood magnitudes. From the perspective of statistics, POT3, as has a larger amount of magnitude values, could be considered to ensure a relatively lower uncertainty caused by sample size. The conservative and also often adopted solution is to give a fair comparison of both acceptable choices of POT2 and POT3 and then make a decision related to the actual requirements of a specific engineering project.

(15) Page 30, line 618: Dot is missing at the end of the sentence.

**Response:**

Thanks for pointing out this omit. It has been added.

(16) Page 31, line 642: Replace "shows" with "show".

**Response:**

Thanks. It has been corrected.

(17) Page 32, line 652: "there is not much difference" looking at Fig. 5 I would say that differences are relatively large for some cases?

**Response:**

Thanks for pointing out this imprecise expression. It has been rephrased in the revised manuscript. Here the statement "there is not much difference" should have been specified to the results of some return periods $T$ (e.g., 30 years) when using the LNO3 and LP3 models for flood estimation.

(18) Page 33, line 672: Replace "if we allowing" with "if we are allowing".

**Response:**

Thanks, the syntax error has been corrected.

(19) Page 33, line 679: Replace "requires" with "require".

**Response:**

Thanks, it has been revised.

(20) Page 36, line 748: Reason for this difference?

**Response:**

Thanks for this good question. Please refer to the response to Specific comment (13) that is very similar to this comment.

(21) Page 36, lines 748-751: What does this conclusion means for the practical application of the FFA?

**Response:**

Thanks for this valuable comment. This conclusion is meant to remind practitioners to pay attention to the impact of not only significant changes in flood observation series but also the physical causes (e.g., climate change) behind the changes when conducting a nonstationary flood return level analysis for future projection. In this study, the climate change, as has been demonstrated previously to cause the significantly decreasing trend in AM floods (Xiong et al., 2014), has exerted an important influence on determining the distribution model and accordingly on return level estimations. Due to the climatic impact, the significantly negative trend in AM series (though has been proven) does not mean a sure result of $x_T^{non-s} < x_T^s$ . $x_T^s$

and $x_T^{non-s}$ are *T*-year return levels derived with stationary and nonstationary models of the same type of distribution, respectively.

To further illustrate the possible different results of $x_T^s$ and $x_T^{non-s}$ derived with AM series under the changing climate, we here show some sketch diagrams in Figure 2 below to give the details in the ENE formula. Pr represents the exceedance probability of a single type of distribution. Figure 2a shows the stationary case $x_T^s$ that satisfies $\Pr(X > x_T^s) = 1/T$. Figure 2b shows the most common result of $x_T^{non-s} < x_T^s$, in which the downtrend in AM series only leads to a slightly left-ward shift in the probability density curve (pdc). This result is easily taken for granted before conducting an actual calculation. However, overlooking the possible changes in the shape of pdc (especially the behavior of upper tail) may lead to the wrong conjecture as exemplified in Figure 2c. Figure 2c gives the special case for $x_T^{non-s} = x_T^s$ where the shape of pdc has changed from year to year, the final accumulation of exceedance probability $\sum_{t=t_0+1}^{t_0+T} \Pr(X > x_T^s) = \sum_{t=t_0+1}^{t_0+T} \Pr(X > x_T^{non-s}) = 1$ results in $x_T^{non-s} = x_T^s$.

[Figure]

Figure2 Sketch diagrams for return level inference with the ENE method in the case of stationarity (a), nonstationarity with a left-ward shift in pdc (b), nonstationarity with both a left-ward shift and a varying shape in pdc.

Reference

Du, T., L, Xiong, C.-Y. Xu, C.J. Gippel, S. Guo, and P. Liu (2015), Return period and risk analysis of nonstationary low-flow series under climate change, J. Hydrol., 527, 234–250, doi:10.1016/j.jhydrol.2015.04.041.

Jiang, C., Xiong, L., Wang, D., Liu, P., Guo, S., and Xu, C.-Y.: Separating the impacts of climate change and human activities on runoff using the Budyko-type equations with time-varying parameters, J. Hydrol., 522, 326-338, doi:10.1016/j.jhydrol.2014.12.060, 2015.

Xiong, L., Jiang, C., and Du, T.: Statistical attribution analysis of the nonstationarity of the annual runoff series of the Weihe River, Water Sci. Technol., 70, 939-946, doi:10.2166/wst.2014.322, 2014.

(22) Page 37, lines 760-763: This is very important conclusion but is it true only for this case study or there is a theoretical background for it?

**Response:**

Thanks a lot for this valuable comment. In this paper, the conclusion "that the return level (RL) derived with AM is more sensitivity to climate change than that with the POT" is obtained in the nonstationary context for the study basin of the Weihe. Due to the diversity of the behaviors of nonstationarity and climate change found at different basins, the conclusion obtained in this basin may not be valid for other basins. However, we would like to give two plausible reasons based on this study, which may be helpful for other practitioners. One reason is that the POT series has been demonstrated to be best described by stationary magnitude and nonstationary frequency, while the AM series (only magnitude) has been optimally fitted by the nonstationary model. This indicates that the AM series, which perhaps include small magnitude samples, may overestimate the impact of climate change on variabilities in floods in the real flood. The other reason may be that the AM sample size is too limited in comparison to that of the POT series, thereby leading to the relatively larger uncertainty.

(23) Page 39, lines 807-810: But this "relatively complicated sampling criteria" still exists and if we compare the POT sampling methodology with the nonstationary approach used in this study I would say that it is even more complicated (than the stationary approach) and it requires additional knowledge?

**Response:**

Thank the referee very much for this valuable comment and good suggestion. In the revised manuscript, we have revised the sentences to clarify why "we suggest that POT series should be warranted more attention in nonstationary flood frequency analysis". It is known that many available studies have been still limited to annual maximum extremes such as AM floods, while POT series have so far not received as much emphasis as the AM series, whether on stationary or nonstationary condition. The most likely reason is the complicated sampling criteria of POT series. We should admit that the flood return level analysis with POT series on nonstationary condition does become more complicated than that under the stationary condition. However, the topic on investigation of changes in POT floods is anyway

unavoidable as the evidence of nonstationarity has appeared not only in the magnitude of extreme events but also in the frequency of the events. So we suggest that POT series should be given more focus in nonstationary flood return level analysis. Please see the response to General comment 2.1 for more explanations.

(24) Page 40, 820-823: What does this means from the practical perspective?

**Response:**

Thanks a lot for this good question. We have shown detailed explanation in the response to Specific comment (21) which is very similar to this comment.

(25) Page 40, lines 830-832: Does this hold for this case study or in general?

**Response:**

Thanks for pointing out the vague statement. This finding was obtained based on the specific hydro-climatic circumstances in the Weihe basin where flood magnitude observations show a significant decreasing trend under the impact of climate change, but did not refer to the general conclusion for other basins. For application in different regions, how climate change would influence design floods should be studied according to the actual situations in those regions. In the revised manuscript, we have rephrased the vague statement.

Thanks again for your professional and valuable comments which greatly improve our research and paper.

With best wishes

Yours sincerely

Lihua Xiong, PhD, Professor

State Key Laboratory of Water Resources and Hydropower Engineering Science

Wuhan University

Wuhan 430072, PR China

E-mail: xionglh@whu.edu.cn

Telephone: +86-13871078660

Fax: +86-27-68773568